# The Biological Origins of Soil Organic Matter in Different Land-Uses in the Highlands of Ethiopia

Dessie Assefa [1,2], Axel Mentler [3], Hans Sandén [2], Boris Rewald [2] and Douglas L. Godbold [2,4,*]

1   Department of Natural Resources Management, Bahir Dar University, Bahir Dar P.O. Box 5501, Ethiopia; dessiegenet@gmail.com
2   Department of Forest and Soil Sciences, Institute of Forest Ecology, University of Natural Resources and Life Sciences, Vienna (BOKU), Peter Jordan-Straße 82, 1190 Vienna, Austria; hans.sanden@boku.ac.at (H.S.); boris.rewald@boku.ac.at (B.R.)
3   Department of Forest and Soil Sciences, Institute of Soil Research, University of Natural Resources and Life Sciences, Vienna (BOKU), Peter Jordan-Straße 82, 1190 Vienna, Austria; axel.mentler@boku.ac.at
4   Department of Landscape Carbon Deposition, Global Change Research Institute, Academy of Sciences of the Czech Republic, Na Sádkách 7, 370 05 Ceske Budejovice, Czech Republic
*   Correspondence: douglas.godbold@boku.ac.at

**Abstract:** In the Ethiopian highlands, clearance of Afromontane dry forest and conversion to crop and grazing land lead to land degradation and loss of soil organic matter (SOM). Eucalyptus is often grown on degraded soils, and this results in the partial recovery of soil carbon stocks. The aim of this work was to assess the biological sources of SOM in this land-use sequence. In top-soils (0–10 cm) of four land-use systems, namely remnant natural forest, eucalyptus plantation, cropland, and grazing land, in the Ethiopian highlands, the origin of SOM was investigated. For this, a sequential extraction method was used, involving a solvent extraction, base hydrolysis, and a subsequent CuO oxidation. In these extracts, biomarkers (molecular proxies) were identified to characterize the SOM of the soil of the four land-uses. Putative lipid monomers of leaf, root, and microbial degradation products suggest that root inputs and microbial inputs dominate in SOM of all the land-uses, except grazing land. The ratios of syringyls, vanillyls, and cinnamyls showed that non-woody angiosperm plants were the predominant source for lignin in eucalyptus, cropland, and grazing land soil. In the soils of the natural forest, lignin originates from both woody angiosperms and woody gymnosperms. Our study shows the importance of root and microbial inputs in the formation of SOM, but also that, in the natural forest, legacies of previous forest cover are present.

**Keywords:** suberin; cutin; CuO oxidation; solvent extraction; base hydrolysis; biomarkers

## 1. Introduction

A heavy seasonal rainfall, combined with a topography of steep slopes, causes the extensive topsoil erosion of agricultural soils worldwide [1], including in the Ethiopian highlands [2]. The mean annual precipitation in the central highlands of Ethiopia is ca. 1600 mm, but, in exceptional years, it can be up to 2000 mm; most of the annual precipitation occurs within the months June to September [3]. Topsoil erosion is exacerbated by land-use change, as much of the highlands has been converted from natural forest to crop or grazing land. As a consequence, land degradation is an ongoing process in the Ethiopian highlands [4,5].

Natural forests and woodlands cover ca. 9.5% of the Amhara region, and about 60% of the total area is used as cropland and grazing land [6,7]. The Afromontane dry forest type is characterized by a high tree species diversity mostly of angiosperm genera, with a few gymnosperms, such as *Podocarpus* sp. Thunb. and *Juniperus* sp. (Hochst) ex. Endl. [8]. These forests are almost exclusively confined to sacred groves associated with a church ("church forests") and are thus semi-protected [9]. Eucalyptus is the dominant exotic

species planted in the highland areas because of its fast growth [10], non-palatability to livestock, multiple uses, and high economic return. Tree-less croplands are cultivated with ploughs for growing grain crops. Grazing lands are used as common lands for herding and are often severely degraded.

The conversion of natural forest to agricultural land-use types results in considerable losses of soil organic matter (SOM) [5,11]. For a number of sites in the Ethiopian highlands, Assefa et al. [5] could show losses of about 75% of the SOM stocks of natural forests after conversion to croplands. In this study, the loss of SOM stocks was principally due to a loss of the SOM-rich surface soil layers. The loss of SOM was from both labile and recalcitrant SOM pools [12]. Moreover, in the Ethiopian highlands, Solomon et al. [11] found SOM losses of 55 to 63%, mainly in labile SOM fractions, but also in stable SOM fractions associated with silt. To counter soil degradation, degraded soils are planted with Eucalyptus. Thirty years after afforestation with Eucalyptus, SOM stocks recovered to nearly 70% of the SOM stocks determined in the natural forest [5]. The loss of SOM also affects other chemical and physical properties of soils, such as soil particle density and soil porosity [13], as well as aggregate stability and soil quality [14].

Soil organic matter is a complex mixture of decomposed residues from plant, animal, and microbial origin [15]. Biomarkers, also known as molecular proxies, are molecules that can be attributed to a particular biological material [16–19]. Previous studies have used biomarkers to show the relative contribution of leaf or needle inputs to root inputs to SOM [20], microbial contributions to SOM [21–24], and also sources of lignin in SOM [25].

Sequential biomarker extraction with the application of chromatographic and spectrometric techniques is one of the most commonly used approaches to detect and quantify the concentration of specific biomarkers [23,26,27]. The sequential biomarker extraction techniques include organic solvent extraction and chemolytic methods, such as base hydrolysis and CuO oxidation [28,29]. Extraction with organic solvents isolates unbound (free) lipids, such as *n*-alkanes, *n*-alkanols, *n*-alkanoic acids, steroids, hopanoids, and other terpenoids [16,23,28]. Solvent extraction thus provides a general overview of biomarkers from plant and microbial sources [30,31]. Ester-bound soil lipids are not extractable with organic solvents, but they can be cleaved from SOM by using chemolytic methods, such as base hydrolysis [26,29]. The products of base hydrolysis have been used to trace root or leaf origin inputs based on suberin- and cutin-derived markers, respectively [18,32]. Extraction with CuO oxidation isolates ether bonds to release mainly lignin-derived phenols and other aromatic compounds [33]. Typically, the lignin-derived phenols (vanillyl, syringyl, and cinnamyl) are used to provide information about whether lignin is derived from angiosperm vs. gymnosperm sources, or woody vs. non-woody tissues [31,34]. Whereas gymnosperm wood contains a definite advantage of vanillyl derivatives (vanillin, acetovanillone, and vanillic acid), angiosperm wood is composed of approximately equal quantities of vanillyls and syringyls (syringaldehyde, acetosyringone, and syringic acid) [32,34]. The ratios of syringyls to vanillyls (S/V) and cinnamyl to vanillyl (C/V) monomers from lignin are widely used to differentiate the relative contributions of major plant taxonomic groups (gymnosperms vs. angiosperms) and tissue type (woody vs. non-woody tissue) [25,34]. The S/V ratio of gymnosperm wood is 0, whereas that of angiosperm wood is >1 [34]. Similarly, a higher C/V than 0 indicates the presence of non-woody tissues as cinnamyl monomers (p-coumaric acid, ferulic acid) is abundant in most herbaceous and "soft" tissues (i.e., leaves, grasses, and needles), but absent in wood [34]. The relative contribution of woody to non-woody angiosperms can also be estimated by using the V:S:C ratio. The ratios of those compounds in lignins are plant- and organ-specific non-woody angiosperms tissues (herbs); however, they hold approximately a 1:1:1 V:S:C ratio [25]. To further improve the detection of lignin sources, the lignin phenol vegetation index (LPVI) was developed [25,35]. The LPVI demarcates ranges for angiosperm and gymnosperm tissues. The wood of gymnosperms has an LPVI of <1, whereas wood of angiosperms is between 67 and 415. Non-woody gymnosperm tissues, such as needles, have an LPVI of 3–27, and non-woody angiosperms tissues, such as the leaves of trees and herbaceous plants, as well

as grasses, have a value of 176–2782. Thus, these ratios have a wide and often overlapping range, but are sufficiently distinct to enable the tracing of sources of lignin [25].

Deforestation for agricultural land expansion and extensive land degradation is a major environmental threat in NW Ethiopia [1,2]. In previous work, for a range of sites, we could show severe loss of SOC after deforestation, but also a recovery of SOC after replanting with eucalyptus [5]. In the work presented in this paper, we have focused on one of these sites, Gelawdios, holding the typical land-use change sequence of conversion of natural forest to cropland or grazing land, and then afforestation with eucalyptus after the soils have degraded through agricultural use. Using biomarkers from a sequential extraction, we compare the land-use types to gain an insight into biological sources of SOM in the different systems and potential legacies. We investigated (1) the importance of fine root inputs for formation of SOM, (2) the contribution of microbial inputs to SOM, and (3) the occurrence of legacies of the past forest cover.

## 2. Materials and Methods

### 2.1. Study Area

The study area is located at an elevation of 2500 m above sea level in an undulating landscape typical for the Ethiopian highlands. The climate is temperate with a dry winter and a warm, wet summer, and is classified as Cwb according to the Köppen–Geiger climate classification [36]. The mean annual rainfall at Gelawdios is 1220 mm, with a unimodal rainy season, and the average annual temperature is 19 °C [8]. The site and the land-use systems chosen represent the typical land-use sequence dominating large parts of the highlands. Natural forest is converted into grassland or cropland, and due to the heavy rainfall in the rainy season and the land topography, the soils erode and become degraded. In an attempt to utilize the degraded soils, large areas are planted with eucalyptus.

The soils at different land-uses are Cambisols [37], but with different degrees of soil degradation [12]. The soils have a clay-to-silty-clay texture [12,38], with less than 10% sand, except for the grazing land, which has 24% sand and less clay [38]. The pH of the soils is similar for all land-uses and is between 6.0 and 6.2 in water and between 5.0 and 5.8 in 0.01 M CaCl$_2$ [12,38]. The water-holding capacity of the soils decreases in the order natural forest > eucalyptus > cropland = grazing land [38].

The study was carried out by using four adjacent land-use systems (natural forest, eucalyptus plantation, cropland, and grazing land) at Gelawdios, Amhara, Northwestern Ethiopia (11°38′25″ N and 37°48′55″ E) (Appendix A Figure A1).

The natural forest is an Afromontane dry forest with a high diversity of indigenous tree species. The Gelawdios forest holds approximately 41 tree species, and the dominant woody species are (sorted by greater basal area first) *Chionanthus mildbraedii* (Gilg & G. Schallenb.), *Euphorbia abyssinica* J.F.Gmel, *Apodytes dimidiate* E.MEY ex ARN, *Schefflera abyssinica* (Hochst. ex A.Rich.) Harms, *Ekebergia capensis* Sparrm., *Albizia schimperiana* Oliv., *Calpurnia aurea* (Aiton) Benth., *Combretum molle* R.Br. ex G.Don and *Croton macrostachyus* Hochst., all of which are angiosperms. The only woody gymnosperms occurring in the forest is *Podocarpus falcatus* (Thunb.) C.N.Page, which occurs in clumps and makes up only a small percentage of the basal area. *Juniperus procera* (Hochst) ex. Endl occurs rarely and is mainly restricted to areas around the Gelawdios church, which is now outside the main forest area. The Gelawdios forest has a density of approximately 6300 trees per hectare above a diameter at breast height (dbh) of 5 cm [5]. Due to the high tree density, undergrowth of grasses and herbs in the forest is minor. Trees cannot be cut in the forest, but the collection of dead wood is allowed.

The eucalyptus (*Eucalyptus globulus* Labill.) stand was established on formerly common grazing land in 1985 and was consecutively thinned to its current density of about 3000 trees per hectare [5]. The understory is dominated by grasses and indigenous shrub species. Trees and deadwood are cut and removed from the forest; leaves are also collected and removed.

The adjacent grazing land and cropland were converted from natural forest approximately in the early 1970s (the exact date is not known). The grazing land is used as

communal grazing lands for herds of animals and it consists of highly degraded, nearly bare ground. Across the grazing land, there are sporadic trees present, mainly *Croton macrostachyus*. The cropland is tree-less, and it is usually cultivated by using an animal-drawn wooden plough to a depth of ca. 15 cm. After harvest, the crop residues are either removed for animal feeding, or the animals are allowed to graze the stubble. The principal crops are "teff" (*Eragrostis tef* (Zucc.) Trotter), wheat (*Triticum aestivum* L.), barley (*Hordeum vulgare* L.), maize (*Zea mays* L.), and sorghum (*Sorghum bicolor* (L.) Moench).

### 2.2. Soil Sampling, C and N Analysis

Soil samples were collected at the end of the wet season in September 2015 from all land-use types. Soil samples were taken from the top 10 cm of the soils from the natural forest, eucalyptus plantation, cropland, and grazing land. The samples were taken at 10 sampling points marked at 50–100 m distance along a transect line. At each sampling point, one sample was taken from the depth of 0–10 cm with a soil corer (6.8 cm in diameter) after removing the litter layer (if present). The samples were air-dried within one week of sampling, and the soil samples were sieved (2 mm) and packed separately in plastic bags. The samples were transported in October 2015 to Vienna for laboratory analysis. In Vienna, subsamples of soils weighing approximately 3–5 g from each sample were dried at 105 °C for 48 h. From each sample, about 200 mg of soil was taken, and total C and N concentrations were determined on a CN elemental analyzer (Truspec CNS, LECO, St. Joseph, MO, USA).

### 2.3. Sequential Chemical Extraction

Sequential chemical extractions (solvent extraction, base hydrolysis, and CuO oxidation) were conducted on soil samples to determine total solvent extracts, bound lipids, and lignin-derived phenols, respectively [32,39]. From the 10 soil samples taken along each transect, equal weights of three, three, and four samples were mixed to give three combined samples for further analysis.

Solvent extraction: The three soil samples (20 g) from each land-use type were first sonicated twice for 15 min, each time with 30 mL double deionized water (DDW) to remove the water-soluble polar compounds. The water-extracted soil residues (~20 g) were then freeze-dried and extracted with organic solvents as follows: samples were sonicated for 15 min with 50 mL of methanol, dichloromethane:methanol (DCM-MeOH; 1:1; *v/v*), and DCM, sequentially. The combined solvent extracts were passed through glass-fiber filters (Whatman GF/A) into a round-bottom flask, concentrated by rotary evaporation, and then completely dried under nitrogen gas ($N_2$) in 2 mL glass vials. The remaining soil samples (non-extractable materials) were air-dried for further analysis.

Base hydrolysis: The air-dried soil residues from solvent extraction were then subject to base hydrolysis to yield ester-linked lipids [26]. Briefly, the residues after solvent extraction were heated at 100 °C for 3 h in Teflon-lined bombs with 20 mL of 1 M methanolic KOH. After cooling, the extracts were acidified to pH 1 with 6 M HCl and filtered through pre-extracted cellulose filters (Fisher P5, 5–10 μm). Again, the soil residues were extracted twice by sonication for 15 min with 30 mL DCM-MeOH (1:1; *v/v*), as described above. The two extracts were combined and then filtered through glass-fiber filters (Whatman GF/A) into round bottom flasks. DDW (50 mL) was added to each extract. Lipids were recovered from the water phase by liquid–liquid extraction in a separation funnel with 50 mL of diethyl ether. Anhydrous $Na_2SO_4$ was added to the combined ether phases to remove any water. The ether extracts were concentrated by rotary evaporation, transferred to 2 mL glass vials, and dried under $N_2$ gas. The remaining soil samples were air-dried for further analysis.

CuO oxidation: This involved the extraction by CuO oxidation, followed the method of Otto et al. [17]. In brief, the base hydrolysis residues were air-dried and further oxidized with CuO to release lignin-derived phenols. Soil residues (~10 g) were extracted with 1 g CuO, 100 mg ammonium iron (II) sulfate hexahydrate [$Fe(NH_4)_2(SO_4)_2 \cdot 6H_2O$], and 15 mL of 2 M NaOH in Teflon-lined bombs at 170 °C for 2.5 h. After heating, the bombs were cooled under running water; the liquids were decanted into Teflon centrifuge tubes (50 mL); and the

residues were washed twice each with 10 mL deionized water, using a magnetic stirrer for 10 min. The combined washings and extracts were centrifuged for 30 min at $1050\times g$ force (Heraeus Megafuge 1.0, Hanau, Germany). The supernatant was decanted into another Teflon centrifuge tube, acidified to pH 1 with 6 M HCl, and kept for 1 h at room temperature in the dark to prevent reactions of cinnamic acids. After centrifugation (at $1050\times g$ force for 30 min), the supernatant was transferred to a separation funnel and liquid–liquid extracted twice with 50 mL diethyl ether. The ether extracts were concentrated by rotary evaporation, transferred to 2 mL glass vials, and dried under $N_2$ gas. During sample preparation, one replicate was lost; thus, *n* = 2.

### 2.4. Derivatization and GC/MS Analysis

Derivatization was conducted according to References [26,32]. In brief, the extracts were re-dissolved, and aliquots (containing ~1 mg extracts) were derivatized for GC/MS analysis. Solvent extracts and CuO oxidation products were each re-dissolved in 500 μL DCM-MeOH (1:1; *v/v*). Aliquots of the extracts (100 μL) were dried under a stream of $N_2$ and then converted to trimethylsilyl (TMS) derivatives by 90 μL N,O-bis-(trimethylsilyl)trifluoroacetamide (BSTFA) and 10 μL pyridine for 3 h at 70 °C. After cooling, 100 μL hexane was added to dilute the extracts. The base hydrolysis products were first methylated by reacting with 600 μL of diazomethane in ether at 37 °C for 1 h, evaporated to dryness under $N_2$, and then silylated with BSTFA and pyridine as described above. Oleic acid (C18:1 alkanoic acid), tetracosane, and ergosterol were derivatized and used as external standards for solvent extracts. Oleic acid methyl ester and vanillic acid were used as external standards for base hydrolysis and CuO oxidation products, respectively. To ensure a linear response, standards ranging from 10 to 1000 ppm were analyzed by using the same procedure as the extracts. GC/MS analysis was performed on an Agilent model 6890N GC coupled to a Hewlett-Packard model 5975 quadrupole mass selective detector. Separation was achieved on an HP5-MS fused silica capillary column (30 m × 0.25 mm internal diameter, 0.25 μm film thickness). The GC operating condition: The temperature was held at 65 °C for 2 min and then increased from 65 to 300 °C at a rate of 6 °C min$^{-1}$ with a final isothermal hold at 300 °C for 20 min. Helium was used as the carrier gas. The samples were injected with a 2:1 split ratio, and the injector temperature was set at 280 °C. The sample (1 μL) was injected with an Agilent 7683B autosampler. The mass spectrometer was operated in the electron impact mode (EI) at 70 eV ionization energy and scanned from 50 to 650 Daltons. Data were acquired and processed with the Chemstation G1701FA software.

Individual compounds were identified by comparing the mass spectra with the National Institute of Standards and Technology library (NIST, version 2.0), Wiley MS library data, and standards. Concentrations of individual compounds (Appendix A Figures A1–A3 and Tables A1–A3) were calculated by comparing the peak area of the compound to that of the standards and were then normalized to the soil carbon contents [40]; partially, sums of major compound classes were calculated. The biomarker concentrations and total yields are expressed as organic C-normalized in μg g$^{-1}$ C or mg g$^{-1}$ C [40].

### 2.5. Calculated Ratios and Indices

2.5.1. Microbial Biomarkers

To determine the relative contribution of plant and microbial residues in the SOM, in the solvent extracts, the sums of different chain lengths of *n*-alkanols, *n*-alkanes, and *n*-alkanoic acids were used. The plant fraction was estimated from $\geq C_{20}$, and microbial from $< C_{20}$. [21,32]. In the base hydrolysis extracts, for the plant fraction, the sums of the long-chain ($\geq C_{20}$) lipids, *n*-alkanols, *n*-alkanes, and *n*-alkanoic acids with even-over-odd preference were used [21,32]. For the microbial fraction, sums of the short-chain alkanes, alkanoic acids and diacids, mid-chain substituted hydroxy alkanoic acids ($C_{16}$, $C_{18}$, and $C_{19}$), branched alkanoic acid (iso-$C_{16}$), and α-alkanoic acids ($C_{16}$–$C_{18}$) were used ($< C_{20}$) [21,32]. To determine the relative inputs of plant materials to microbial materials, a ratio was calculated ($< C_{20}$:$\geq C_{20}$).

### 2.5.2. Suberin and Cutin Monomers

As uncertainty exists about the cutin or suberin origin of a number of compounds [18,28], sums of suberin monomers were calculated by using two methods. Suberin 1 was calculated as the sum of ω-hydroxyalkanoic acids ($C_{20}$–$C_{30}$) + α,ω-alkanedioic acids ($C_{20}$), and suberin 2 as the sum of ω-hydroxyalkanoic acids ($C_{16}$–$C_{30}$) + α,ω-alkanedioic acids ($C_{20}$). Similarly, cutin 1 was calculated as the of $C_{16}$ mono and dihydroxy acids and diacids, and cutin 2 as the sum of $C_{16}$ mono and dihydroxy acids and diacids + $C_{18}$ trihydroxy acids. Suberin 1 and cutin 1 are the sum of monomers that are most likely from suberin or cutin, respectively, whereas suberin 2 and cutin 2 contain monomers of which there is debate about their origin [18,28]. The suberin-to-cutin ratio was calculated by using suberin 1 and cutin 1, and suberin 2 and cutin 2, respectively.

### 2.5.3. Lignin-Derived Phenols

To determine major plant taxonomic groups (gymnosperms vs. angiosperms) and tissue type (woody vs. non-woody tissue) from lignin-derived phenols, ratios of syringyl to vanillyl (S/V) and cinnamyl to vanillyl (C/V) monomers, as identified in the CuO oxidation extracts, were calculated [34]. For this calculation, the compounds *p*-coumaric acid and ferulic acid (cinnamyls, C), vanillin/vanillaldehyde, acetovanillone and vanillic acid (vanillyls, V), and syringaldehyde, acetosyringone and syringic acid (syringyls, S) were used [34].

The relative contribution of woody and non-woody angiosperms was estimated by using the vanillyl-to-syringyl-to-cinnamyl (V:S:C) ratio [34].

The lignin phenol vegetation index (LPVI) was calculated based on Reference [35].

$$\text{LPVI} = [\{S(S + 1)/(V + 1) + 1\} \times \{C(C + 1)/(V + 1) + 1\}] \tag{1}$$

where V, S, and C are expressed as % of total VSC.

### 2.6. Statistical Analysis

Values for the compounds shown in the tables and tables in the appendix are means and standard errors (SE; *n* = 3). Data for selected compounds and summed compound groups were checked for normality (Shapiro–Wilk) and equal variance (Brown–Forsythe). All datasets met the conditions of normality and were analyzed by a parametric one-way ANOVA test, with a post hoc Tukey test. The analysis was carried out by using the program Sigmastat 4.0 (Systat GmbH, Frankfurt, Germany).

## 3. Results

### 3.1. Soil C and N Content, and Yields of Sequential Extractions

The percentage of carbon in the top 10 cm of the soil of the different land-use types was significantly higher in the natural forest and the eucalyptus plantation when compared to the cropland and grassland (Figure 1). The soil of the natural forest had an approximately 4.4- and 3.7-times greater concentration of C compared to the grassland and the cropland, respectively, and a 3-times greater concentration compared to the eucalyptus plantation. The C concentration of the soil at the eucalyptus stand was only ca. 25% higher than that of the grassland and the cropland. Soil N concentrations followed a trend similar to the C contents.

The normalized yield of identified biomarkers per unit soil C differed between the extraction methods, i.e., the solvent extract, base hydrolysis, and CuO oxidation (Table 1). Among the sequential extraction steps, the highest normalized yield was obtained from the base hydrolysis products. The normalized yields also differed between the soils of the different land-use types within an extraction method. In the solvent extraction, the normalized yield was significantly greater from eucalyptus soil compared to cropland and grazing land soils. The normalized yield of base hydrolysis products was significantly less from the natural forest and eucalyptus soils compared to the grazing land. In contrast, the normalized yield from CuO oxidation did not differ between the soils of the land-use types.

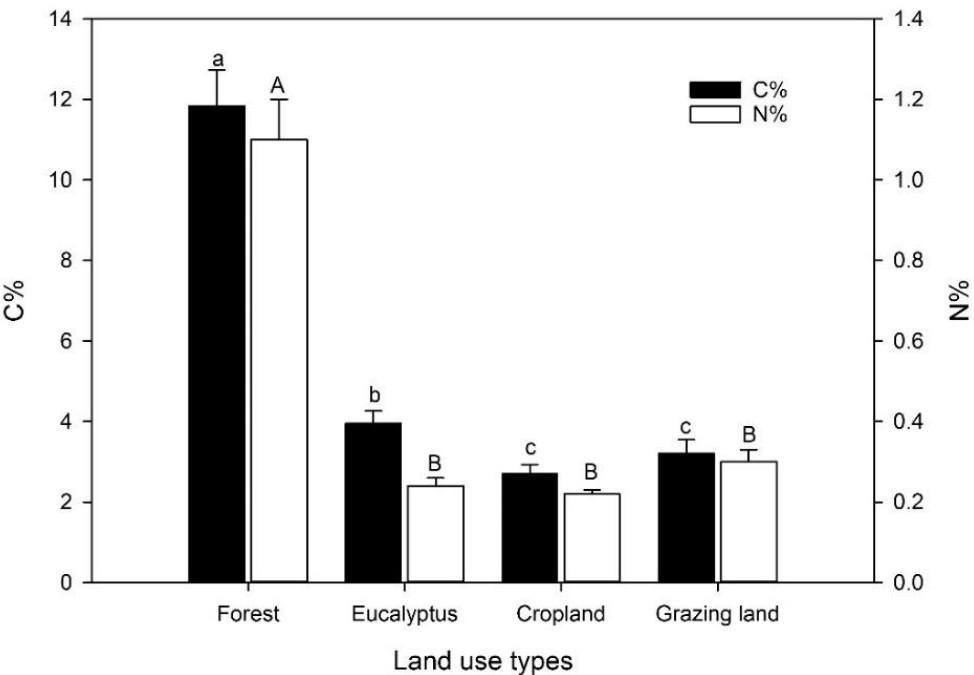

**Figure 1.** Carbon (C) and nitrogen (N) contents (%) at 0–10 cm soil depth in four land-use systems at Gelawdios, Ethiopia (mean ± SE, *n* = 10). Different lower-case letters indicate significant differences in soil C content; upper-case letters indicate significant differences in N content between land-use types. Note: 10-fold difference in axis scales between C and N contents.

**Table 1.** Normalized yields (mg g$^{-1}$ C) of the three extraction methods from soils of different land-use systems in Gelawdios, Ethiopia. Shown are the mean and range (in parentheses) for the natural forest and mean ± SE (*n* = 3) for the other land-uses. Different letters indicate a significant difference between land-uses ($p \leq 0.05$).

| Extraction | Forest | Eucalyptus | Cropland | Grazing Land |
|---|---|---|---|---|
| Solvent | 5.6 ± 0.7 ab | 8.2 ± 0.9 b | 3.3 ± 0.5 a | 3.2 ± 0.4 a |
| Base hydrolysis | 30.0 ± 3.6 a | 82.0 ± 1.0 b | 97.4 ± 4.4 bc | 111.3 ± 8.2 c |
| CuO oxidation | 36.0 (33.2–38.7) | 21.3 ± 2.1 a | 21.4 ± 1.0 a | 21.4 ± 4.9 a |

*3.2. Solvent-Extractable Compounds*

For the solvent-extractable compounds, the GC–MS total ion chromatogram of the major components in the silylated solvent extracts is shown in Appendix A Figure A2, and all individual compounds are listed in Appendix A Table A1. The major chemical compounds detected in the solvent extract include a series of aliphatic lipids (*n*-alkanols, *n*-alkanes, and *n*-alkanoic acids), carbohydrates, monoacylglycerides, steroids, and terpenoids (Table 2). Sugars, followed by aliphatic lipids, were the largest component of the total solvent extract in all land-uses, except eucalyptus. In eucalyptus soil, the concentration of steroids and terpenoids exceeded the concentration of both sugars and total aliphatic lipids.

**Table 2.** Compounds, their sums, and their ratios, as identified in the solvent (s) extracts of soil samples of different land-use systems in Gelawdios, Ethiopia (mean $\pm$ SE, $n$ = 3). All values are in ($\mu$g g$^{-1}$ C). Different letters indicate a significant difference between land-uses ($p \leq 0.05$).

| Compound [1] | Forest | Eucalyptus | Cropland | Grazing Land |
|---|---|---|---|---|
| $n$-Alkanols ($C_{14}$–$C_{30}$) | 324 $\pm$ 171 a | 520 $\pm$ 212 a | 250 $\pm$ 86 a | 356 $\pm$ 103 a |
| $n$-Alkanes ($C_{17}$–$C_{31}$) | 222 $\pm$ 71 ab | 391 $\pm$ 78 a | 139 $\pm$ 66 ab | 58 $\pm$ 15 b |
| $n$-Alkanoic acids ($C_9$–$C_{26}$) | 448 $\pm$ 184 b | 1129 $\pm$ 128 a | 207 $\pm$ 49 b | 272 $\pm$ 31 b |
| Sum | 995 $\pm$ 288 ab | 2040 $\pm$ 400 b | 596 $\pm$ 198 a | 686 $\pm$ 120 a |
| Sum < $C_{20}$ | 172 $\pm$ 55 a | 1269 $\pm$ 221 b | 205 $\pm$ 60 a | 205 $\pm$ 59 a |
| Sum $\geq$ $C_{20}$ | 823 $\pm$ 236 a | 771 $\pm$ 180 a | 391 $\pm$ 140 a | 482 $\pm$ 81 a |
| <$C_{20}$:$\geq$$C_{20}$ ratio (s) | 0.21 $\pm$ 0.01 a | 1.69 $\pm$ 0.09 c | 0.55 $\pm$ 0.07 b | 0.43 $\pm$ 0.10 ab |
| Monoacylglycerides ($C_{19}$–$C_{21}$) | 52 $\pm$ 26 a | 260 $\pm$ 19 b | 134 $\pm$ 46 ab | 84 $\pm$ 2 a |
| Sugars | 3936 $\pm$ 703 a | 2429 $\pm$ 256 ab | 1820 $\pm$ 229 b | 1894 $\pm$ 192 b |
| Steroids and Terpenoids | 622 $\pm$ 267 a | 3461 $\pm$ 341 b | 729 $\pm$ 80 a | 616 $\pm$ 99 a |

[1] All polar compounds were identified as their trimethylsilyl (TMS) derivatives.

The concentration of $n$-alkanols in the range of $C_{20}$–$C_{30}$ and $n$-alkanoic acids in the range of $C_9$–$C_{26}$ showed a significant even-over-odd dominance of C-chain lengths. Among $n$-alkanols, only one compound, 1-Tricosanol, had an odd C chain length; among $n$-alkanoic acids, two compounds, nonanoic acid and tricosanoic acid, had uneven C chain lengths. In contrast, the $n$-alkanes in the range of $C_{17}$–$C_{31}$ had an odd-over-even dominance, principally induced through the concentrations of $n$-heptadecane and $n$-heptacosane (Appendix A Table A1). aliphatic lipids were significantly greater in eucalyptus soil compared to the grassland and cropland soils, but not compared to the forest soil (Table 2). The aliphatic lipids with a C chain length less than 20 (<$C_{20}$) were significantly higher in the eucalyptus soil, but no differences were found for chain lengths of 20 and above ($\geq$$C_{20}$). In the natural forest soil, the sum of <$C_{20}$ was significantly less than the sum of $\geq$$C_{20}$; the opposite was found in the eucalyptus soil. As a consequence, the <$C_{20}$:$\geq$$C_{20}$ ratio in eucalyptus soil was significantly greater than in all other land-use types. For forest soil, this ratio was significantly less compared to cropland soils.

Four sugars (glucose, mannose, sucrose, and trehalose) were identified in the solvent extracts (Appendix A Table A1). Trehalose composed 87% (eucalyptus) to 99% (natural forest) of the total sugars. The sum of all sugar concentrations was significantly greater in the natural forest soil compared to the cropland and grassland; this difference was driven by the concentration of trehalose, which was significantly more abundant in natural forest soil compared to all other land-uses (Appendix A Table A1). The solvent extracts also included four monoacylglycerides, three of them being $C_{21}$ compounds.

Steroids and terpenoids were ca. 5-times more abundant in eucalyptus soil compared to the other land-uses (Appendix A Table A2). The eucalyptus soil contained higher amounts of a number of specific steroids and terpenoids (Appendix A Table A1). Sesquiterpenes of trans-farnesol and globulol combined contributed 38% of total steroids; other sesquiterpenes detected included aromadendrene, $\gamma$-elemene, and ledol. These sesquiterpenes were found in minor amounts in the cropland soil, but not in the other land-uses. The eucalyptus soil also had high concentrations of triterpenoids, erythrodiol, and oleanolic acid. Ergosterol was only found in the forest and eucalyptus soil. The detected phytosterols included campesterol, stigmasterol, $\beta$-sitosterol, and sitosterone; these compounds contributed over 81–96% of total steroids in the forest, cropland, and grazing land, but only 19% in eucalyptus soil (Appendix A Table A1).

*3.3. Base Hydrolysis Extractable Compounds*

The major products identified after base hydrolysis included a series of aliphatic lipids ($n$-alkanols, $n$-alkanoic acids, mid-chain substituted and branched acids, $\omega$-hydroxyalkanoic acids, $\alpha$-hydroxyalkanoic acids, $\alpha$,$\omega$-alkanedioic acids, and glycerides), with a lesser contribution from benzyls and phenols (Table 3), and one steroid (Appendix A Table A2).

The compounds that were identified following base hydrolysis are listed in Appendix A Table A2, and examples of the GC–MS chromatograms are shown in Appendix A Figure A3. Aliphatic lipids showed a similar pattern and represented more than 60% of the total base extracts; *n*-alkanoic acids in the range of $C_{14}$–$C_{30}$ were the dominant fraction. The *n*-alkanoic acids with a chain length less than 20 ($<C_{20}$) greatly exceeded the sum of the *n*-alkanoic acids with a chain length greater than 20 ($\geq C_{20}$) by a factor of 8–15 (Table 3). For both chain-length classes, there was a strong even-over-odd preference (Appendix A Table A2). The $\alpha,\omega$-alkanedioic acids, mid-chain substituted hydroxyalkanoic acids, $\alpha$-alkanoic acids, $\omega$-alkanoic acids, and glycerides were detected in the soils of each land-use in similar concentrations. Branched alkanoic acids (iso-$C_{16}$) were not detected in grazing land soil (Table 3), but they were in the soils of the other land-uses. Base hydrolysis also cleaved monoacylglycerides ($C_{19}$) in minor concentrations (238–487 $\mu g\ g^{-1}$ C) in all land-use types besides grazing land. One organophosphate of $C_{19}$ was also detected in soil of all land-uses, with significantly greater concentrations detected in cropland and grazing land soils. Benzyls and phenols ranged from 0.4 to 2.3 mg $g^{-1}$ C (Table 3).

**Table 3.** Compounds, their sums, and their ratios, as identified from base hydrolysis (bh) of soil samples in different land-use systems in Gelawdios, Ethiopia (mean $\pm$ SE, *n* = 3). All values are in $\mu g\ g^{-1}$ C or mg $g^{-1}$ C, as shown. Different letters indicate a significant difference between land-uses ($p \leq 0.05$).

| Compound | Forest | Eucalyptus | Cropland | Grazing Land |
|---|---|---|---|---|
| | | $\mu g\ g^{-1}$ C | | |
| *n*-Alkanols ($C_{16}$–$C_{28}$) [1] | 104 $\pm$ 21 a | 265 $\pm$ 12 a | 314 $\pm$ 21 a | 627 $\pm$ 362 a |
| *n*-Alkanoic acids ($C_{14}$–$C_{30}$) [2] | 4788 $\pm$ 1180 a | 9998 $\pm$ 179 b | 11063 $\pm$ 452 b | 12157 $\pm$ 1173 b |
| Branched alkanoic acids (iso-$C_{16}$) | 11 a | 38 a | 16 a | nd |
| $\alpha$-Alkanoic acids ($C_{16}$–$C_{25}$) 3 | 364 $\pm$ 60 a | 515 $\pm$ 124 a | 576 $\pm$ 100 a | 571 $\pm$ 357 a |
| Sum $<C_{20}$ | 4901 $\pm$ 1113 a | 9799 $\pm$ 107 b | 10789 $\pm$ 357 b | 11664 $\pm$ 1317 b |
| Sum $\geq C_{20}$ | 309 $\pm$ 43 a | 873 $\pm$ 79 bc | 994 $\pm$ 237 bc | 1616 $\pm$ 447 c |
| $<C_{20}$:$\geq C_{20}$ ratio (bh) | 15.9 $\pm$ 3.4 a | 11.4 $\pm$ 1.0 a | 11.9 $\pm$ 2.2 a | 7.9 $\pm$ 1.2 a |
| $\alpha,\omega$-Alkanedioic acids ($C_4$–$C_{20}$) 3 | 342 $\pm$ 49 a | 642 $\pm$ 41 a | 589 $\pm$ 125 a | 678 $\pm$ 69 a |
| $\omega$-Hydroxyalkanoic acids ($C_{16}$–$C_{30}$) 3 | 736 $\pm$ 31 a | 1105 $\pm$ 26 b | 1344 $\pm$ 30 b | 298 $\pm$ 141 c |
| Mid-chain substituted hydroxy acids | 227 $\pm$ 111 a | 592 $\pm$ 122 a | 422 $\pm$ 160 a | 309 $\pm$ 80 a |
| Monoacylglycerides ($C_{19}$) | 238 $\pm$ 59 | 487 $\pm$ 128 | 432 $\pm$ 177 | nd |
| Benzyles and phenols | 428 $\pm$ 53 | 1233 $\pm$ 57 | 1390 $\pm$ 200 | 2303 $\pm$ 493 |
| | | mg $g^{-1}$ C | | |
| Organophosphates | 1.5 $\pm$ 0.2 a | 4.5 $\pm$ 0.1 b | 5.6 $\pm$ 0.3 bc | 6.8 $\pm$ 0.6 c |

[1] *n*-Alcohols, and ß-Sitosterol were identified as TMS ethers. [2] Alkanoic acids were identified as methyl esters and hydroxyacids as methyl esters/TMS ethers. [3] Phenolic acids were identified as methyl esters/TMS ethers. nd = not detected.

The concentration of the putative suberin monomers (Table 4), suberin 1 or suberin 2, was significantly lower in the natural forest compared to the eucalyptus and cropland soils, but not the grazing land where values were (significantly) lower. For the sum of putative cutin monomers calculated as cutin 1 and cutin 2, there was no significant difference between land-uses. Both suberin/cutin ratios were similar in soils of all land-uses, indicating an excess of suberin, except for grazing land; however, there were no significant differences between the land-uses.

**Table 4.** Sums of suberin or cutin monomers (µg g$^{-1}$ C) identified from base hydrolysis of soil samples in different land-use systems in Gelawdios, Ethiopia (mean ± SE, $n$ = 3). Different letters indicate a significant difference between land-uses ($p \leq 0.05$).

| Suberin and Cutin Monomers | Forest | Eucalyptus | Cropland | Grazing Land |
|---|---|---|---|---|
| Suberin 1 | 653 ± 56 a | 1087 ± 63 b | 1247 ± 15 b | 322 ± 137 c |
| Suberin 2 | 792 ± 42 a | 1267 ± 27 b | 1488 ± 36 b | 456 ± 129 a |
| Cutin 1 | 126 ± 40 a | 268 ± 96 a | 305 ± 90 a | 223 ± 17 a |
| Cutin 2 | 158 ± 71 a | 359 ± 88 a | 344 ± 109 a | 223 ± 17 a |
| Suberin/Cutin ratio 1 | 6.1 ± 1.4 a | 6.3 ± 3.1 a | 5.3 ± 2.2 a | 1.0 ± 0.4 a |
| Suberin/Cutin ratio 2 | 7.1 ± 2.5 a | 3.9 ± 0.9 a | 6.1 ± 2.8 a | 1.5 ± 0.6 a |

Suberin 1 = ω-hydroxyalkanoic acids ($C_{20}$–$C_{30}$) + α,ω-alkanedioic acids ($C_{20}$). Suberin 2 = ω-hydroxyalkanoic acids ($C_{16}$–$C_{30}$) + α,ω-alkanedioic acids ($C_{20}$). Cutin 1 = $C_{16}$ mono and dihydroxy acids and diacids. Cutin 2 = $C_{16}$ mono and dihydroxy acids and diacids + $C_{18}$ trihydroxy acids.

### 3.4. CuO Oxidation Extractable Compounds

The CuO oxidation of soils released benzyls, lignin-derived phenols, lipid-derived carboxylic acids (short-chain alkanedioic and hydroxy acids), and cutin-derived products, as well as compounds derived from polysaccharides, proteins, and tannins (Appendix A Figure A4 and Table A3). In forest soil, the total concentration of identified C-normalized CuO products was almost double the levels determined in the soils of other land-uses, but it was not statistically different. A trend of higher values in soil from the natural forest was seen for the majority of the compounds determined. In addition to the eight major lignin-derived phenols, nine benzyls, and three other phenols were identified (Appendix A Table A3).

The syringyl to cinnamyl (S/V) and cinnamyl to vanillyl (C/V) ratios were calculated for the sums of the lignin-specific phenols (Table 5). Both the S/V and C/V ratios were lower in natural forest soil than in the other land-use systems, which had similar values. Similarly, the V:S:C ratio differed strongly in natural forest soil, with a ratio of 4:2:1, compared to the other land-uses, which all had a ratio close to 1:1:1. The lignin phenol vegetation index (LPVI) was the lowest in natural forest and highest in grazing land soils. The eucalyptus and cropland soils had a similar intermediate value.

**Table 5.** Syringyl-to-cinnamyl ratio (S/V), cinnamyl-to-vanillyl ratio (S/V), vanillyl-to-syringyl-to-cinnamyl ratio (V:S:C), and the lignin phenol vegetation index (LPVI) in soil of four land-use systems at Gelawdios, Ethiopia. For S/V and C/V ratios and for the LPVI, mean and range are shown.

| Index | Forest | Eucalyptus | Cropland | Grazing Land |
|---|---|---|---|---|
| S/V | 0.6 (0.5–0.7) | 1.1 (0.7–1.4) | 1.0 (1.0–1.0) | 1.4 (1.2–1.6) |
| C/V | 0.3 (0.2–0.3) | 0.8 (0.5–1.0) | 0.9 (0.6–1.0) | 1.3 (0.9–1.9) |
| V:S:C | 4:2:1 | 1:1:1 | 1:1:1 | 1:1:1 |
| LPVI | 70 (50–90) | 992 (301–1674) | 880 (548–1087) | 2080 (1323–2941) |

A plot of the S/V and C/V ratios (Figure 2) showed a clear separation of the land-use systems. The eucalyptus, cropland, and grazing land were grouped in the range of the non-woody angiosperms, with a clear difference between eucalyptus and cropland to the grazing land. The natural forest also showed a clear separation from the other land-use system and separated toward the range of gymnosperm wood and angiosperm wood.

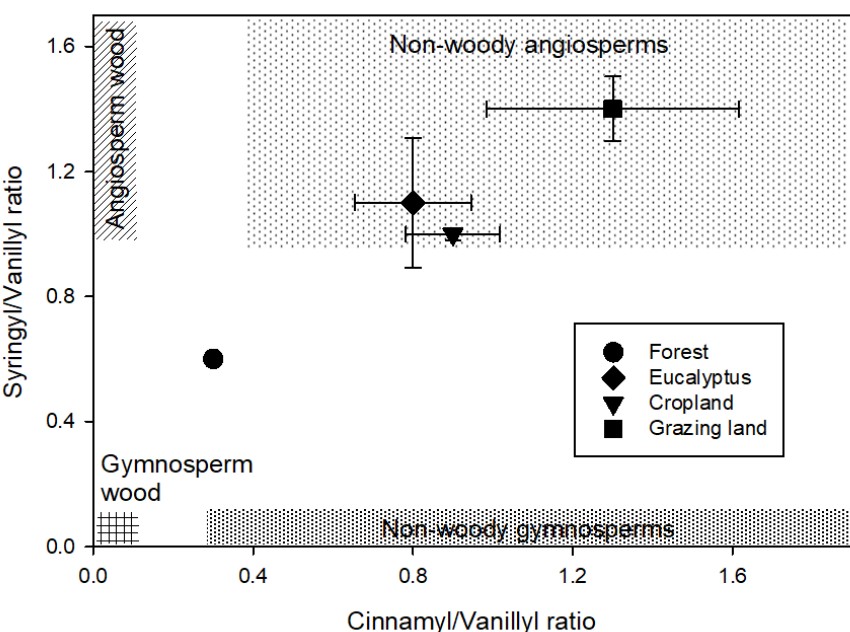

**Figure 2.** Lignin source parameters of the monomeric syringyl/vanillyl (S/V) and cinnamyl/vanillyl (C/V) phenol ratios among four land-use systems at Gelawdios, Ethiopia. Patterned areas indicate the expected S/V to C/V relationships of four vegetation types according to Reference [35]. Shown are mean $\pm$ SE.

## 4. Discussion

### 4.1. Above- and Belowground Inputs

Soil organic matter (SOM) is formed through the inputs of above- and belowground plant materials and from microbial detritus [15]. At the Galawdios site, considerable losses of SOM after land-use change have also occurred [5]. In the natural forest, there is substantial necromass input from both aboveground and belowground, whereas, in the other land-uses, the aboveground inputs are small, due to biomass removal [5]. In the natural forest, belowground C inputs from fine roots were almost twice those in the eucalyptus plantation and 8–10 times those in the grassland and cropland [41]. In the natural forest, the amounts of necromass inputs from leaf fall and fine roots were similar [38].

In all land-uses, except for grazing land, the levels of root-derived suberin biomarkers greatly exceeded those of leaf-derived cutin molecular proxies. There is still considerable debate about the cutin or suberin origin of a number of compounds [18,42]; thus, we calculated both a conservative suberin-to-cutin ratio based on compounds that, with a degree of certainty, can be assigned at either suberin to cutin derivatives, and a suberin-to-cutin ratio with a greater degree of uncertainty about the origin of the proxies. The more conservative suberin biomarkers include $\omega$-hydroxyalkanoic acid ($\geq C_{20}$) and $\alpha,\omega$-alkanoic acids ($C_{16}$–$C_{24}$). The more conservative cutin biomarkers include compounds of $C_{16}$ mono- and dihydroxy acids and diacids [42]. Both ratios show a dominance of suberin, suggesting that, in the studied ecosystem, root inputs play a more important role in the formation of SOM than leaf inputs. However, as discussed above, only in the natural forest are the above- and belowground biomass inputs similar. In the eucalyptus plantation, leaves are raked and removed; in the cropland, all crop residues are harvested for either cattle fodder or fuel; and the grazing land is very heavily grazed, removing almost all aboveground biomass, thus explaining the dominance of root-derived biomarkers in these land-uses. In the natural forest with a more balanced input of above- and belowground necromass, the difference still suggests that roots are more important for the formation of SOM than aboveground inputs. A greater importance of belowground inputs compared to aboveground inputs for SOM formation has been shown repeatedly [43–45]. However, there is evidence to suggest that the degradation of suberin biomarkers is slower than that of cutin biomarkers [20,45],

but also that the plant source of the material is important [44]. Hamer et al. [42] showed that grass-derived lipids had a faster turnover than those derived from forest vegetation, and this may explain the low levels of suberin biomarkers found in the grazing land at Gelawdios. In addition, as forest-derived monomers are preserved in the soil for more than 50 years, the high suberin content in cropland may also be, in part, a legacy of the former forest. Particularly, the ω-hydroxy carboxylic acids and α,ω-alkanedioic acids of forest origin may have been stabilized in the soils by bonding to soil minerals. All of the soils have a high clay content [12].

### 4.2. Microbial Inputs

Soil organic matter is considered to originate from both plant and microbial origins [15,43]. Long-chain ($\geq C_{20}$) lipids, such as *n*-alkanols, *n*-alkanes, and *n*-alkanoic acids, with even-over-odd preference are considered to be typical constituents of epicuticular and associated waxes of higher plants [22,32,40]. The short-chain alkanes, alkanoic acids and diacids, mid-chain substituted hydroxy alkanoic acids ($C_{16}$, $C_{18}$, and $C_{19}$), and branched alkanoic acid (iso-$C_{16}$) are considered to be derived from microorganisms [22,40]. However, there is evidence showing that long-chain ($\geq C_{20}$) lipids may also be derived from fungi [19,46]. In the solvent extracts, in all land-uses, except for eucalyptus, the $\geq C_{20}$ compounds exceeded the $< C_{20}$ compounds, suggesting that, particularly in the natural forest, inputs of plant origin dominate this fraction of SOM. The solvent extracts represent a less tight bound and possibly younger fraction of compounds in the soils. However, two other solvent extracted compounds, ergosterol and trehalose, show the presence of microbial activity. Ergosterol, a specific biomarker for living fungal biomass [47], was detected in the forest and eucalyptus soil but not in the cropland and grazing land soil (Appendix A Table A1). The stress protectant trehalose is attributed to fungi, bacteria, and insects, but it is only rarely found in plants [16,48]. The concentration of ergosterol in eucalyptus soil is about 2.6 times that of natural forest soil (Appendix A Table A1) and reflects the ectomycorrhizal status of eucalyptus against the arbuscular mycorrhizal–dominated natural forest [49]. As ergosterol was not detected in cropland and grazing land, microbial inputs associated with the mid-chain alkanoic acids are more likely from bacteria. In the base extracts, which represent the bound lipids and accumulation in the SOM, the concentration of $< C_{20}$ compounds greatly exceeds that of the $\geq C_{20}$ compounds, with no significant difference in the ratio of $< C_{20}:\geq C_{20}$ (bh) between the land-use systems. Thus, indicating the importance of microbial residues in the longer-term SOM pools of all studied land-use systems.

### 4.3. Angiosperms and Gymnosperms

The foot print of eucalyptus can clearly be seen in the solvent extracts, in which the sesquiterpenes, aromadendrene, γ-elemene, ledol, globulol, and farnesol were detected in high concentrations. These compounds are found in a variety of aromatic plants and reported as common constituents of essential oils of eucalyptus [50,51]. Triterpenoids of the ß-amyrin, α-amyrin, lupeol, erythrodiol, and oleanolic acid type are also typical biomarkers for angiosperms [17,40,52]. They were found in large quantities in the eucalyptus soil (Table 1). Particularly, erythrodiol and oleanolic acid are major components in the essential oils of eucalyptus leaves [52].

CuO oxidation was used to release lignin-specific phenols (vanillyl, syringyl, and cinnamyl phenols) from lignin polymers [53]. The S/V ratio of gymnosperm wood and non-woody tissues is very low, due to the absence of syringyls compared to that of angiosperm wood and non-woody tissues [54]. In the different land-uses, the S/V values in soils ranged from 0.61 (natural forest) to 1.4 (grazing land). In another investigation carried out in the Southwestern Ethiopia highlands, the soil of a natural forest with dominance of *Olea africana*, *Syzygium guineense*, *Cordia africana*, and *Croton macrostachys* had a S/V ratio of 0.84. In a similar *Olea* and *Cordia* forest, and also in a forest with a presence of the gymnosperm *Podocarpus falcutus*, a similar value of 0.85 was found [11]. However, in an adjacent gymnosperm *Cupressus* sp. Plantation, the soil had a lower value (0.22). Other studies have also reported lower S/V

values for gymnosperm-dominated species. For example, Li et al. [55] reported a low S/V value (0.23) under pine-dominated forests and 0.84 for farmland soils. The C/V ratio indicates the presence of residues from non-woody tissues of both gymnosperms and angiosperms, as cinnamyls are not present in wood [54]. The C/V ratios were different in all land-use soil samples, again being highest in grassland soil (1.3), followed by cropland (0.9), eucalyptus soil (0.8), and forest soil (0.3). This indicates the presence of material from non-woody tissues in SOM of all land-uses. In cropland, grazing land, and eucalyptus soil, this clearly indicates that non-woody angiosperms, such as grass and herb species, are the predominant sources of lignin. In contrast, in the natural forest ecosystem, the low C/V ratio suggests that both non-woody tissues and wood are important for SOM formation. The V:S:C ratio also supports the major contribution of non-woody angiosperms to SOM in the eucalyptus, cropland, and grazing land soils. Non-woody angiosperms have a 1:1:1 V:S:C ratio [53]. The lignin-derived phenols detected in our samples exhibited a V:S:C ratio of 4:2:1 in natural forest soil and 1:1:1 in the eucalyptus, cropland, and grazing land soils. In a plot of S/V ratio against the C/V ratio [56], a clear separation of the natural forest from the other land-uses was also found. For the natural forest, the lower S/V and C/V values reflect a mixture of both angiosperm and gymnosperm sources. To improve the detection of lignin sources, the lignin phenol vegetation index (LPVI) was developed by Tareq et al. [25]. Tareq et al. [35] calculated the LPVI of 1 for gymnosperm wood, 3–27 for non-woody gymnosperm tissues, 67–415 for angiosperm wood, and 176–2782 for non-woody angiosperm tissues. In the present study, the LPVI value of the natural forest (70) is in the lowest range of angiosperm wood, and below that of angiosperm non-woody tissues. In contrast, soil samples from eucalyptus (992), cropland (880), and grazing land (2080) show characteristics for non-woody angiosperm tissues, and this is consistent with plots of S/V vs. C/V ratios. Thus, all the indices, namely the LPVI, the plots of the S/V ratio against the C/V ratio, and S/V ratio alone, indicate that, in the natural forest, the angiosperm signal might be being lowered by the presence of material from gymnosperms. The exact historical composition of the natural forest is not known, but it was highly likely to be a typical Olive–Juniper-Podocarpus composition. Both *Podocarpus* and *Juniperus* are gymnosperm genera. Today, *Podocarpus falcatus* and *Olea africana* species still occur in the natural forest, whereas *Juniperus procera* is restricted to the surrounding area and compounds of the Gelawdios church. *Juniperus procera* wood is highly valued [57]; much has been harvested, but the regeneration rate is low.

## 5. Conclusions

By using the sequential extraction technique, clearly information about the sources of material for the formation of SOM can be gained. However, there are a number of uncertainties. The composition of leaf and root lipids differs between different vegetation types [42] and between tree species [18], thus affecting the inputs of biomarkers. The natural forest at Gelawdios has over 40 tree species; thus, determining the lipids in the vegetation in the natural forest was beyond the scope of this investigation. Similarly, the relative turnover of biomarkers in soils, a factor that affects all such studies, remains unknown [19]. For comparability, the yields of compounds are normalized against the soil C content, making the comparison of soils with greatly different C contents difficult, and creating differences in the normalized yields between land-uses. For natural forest soil, the normalized yield in the base hydrolysis extract was only 27–37% of that extracted from the other land-uses. However, for the CuO oxidation, the normalized yield for the natural forest soil was 40% higher than the other land-uses. This can be explained by a high concentration of lignin degradation products in the SOM of the natural forest. However, the results clearly show the importance of root inputs for the formation of SOM in all studied land-use types in NW Ethiopia. In addition, in potentially older SOM fractions, microbial inputs are also important in all land-uses. The SOM in the natural forest soil has a strong signal for woody materials from both angiosperms and gymnosperms, and suggest inputs of woody debris are important. This has implications for the restoration of SOM, particularly in eucalyptus,

where trees are cut and most of the woody debris is removed as fuel wood, and in natural forest, where dead trees are legally removed from protected church forests.

**Author Contributions:** D.A., D.L.G., B.R. and H.S. devised the field design and carried out sampling. D.A. carried out all of the analytical work, and together with A.M. carried out the sequential extraction, and peak identification. D.A. and D.L.G. carried out data analysis and co-wrote the original and final manuscript. B.R. and H.S. critically revised drafts of the manuscript. All authors jointly revised and approved the final version of the manuscript. All authors have read and agreed to the published version of the manuscript.

**Funding:** This work was carried out within the project "Carbon storage and soil biodiversity in forest landscapes in Ethiopia: Knowledge base and participatory management (Carbo-part)" and was funded by the Austrian Ministry of Agriculture, Forestry, Environment, and Water Management under grant agreement no. BMLFUW-UW.1.3.2/0122-V/4/2013. DLG was also supported by the Ministry of Education, Youth and Sports of CR within the National Sustainability Program NPU I, grant No. LO1415. HS and BR were partially supported by the University of Natural Resources and Life Sciences, Vienna.

**Data Availability Statement:** Raw data is available on request from the corresponding author.

**Acknowledgments:** We thank Astrid Hobel and Marcel Hirsch for their continuous assistance during laboratory analysis. We would like also provide our special thanks to Andreas Kitzler (Agilent Technologies, Austria) for appropriation of instruments, technical support, and method adaptation.

**Conflicts of Interest:** The authors declare no conflict of interest.

**Appendix A**

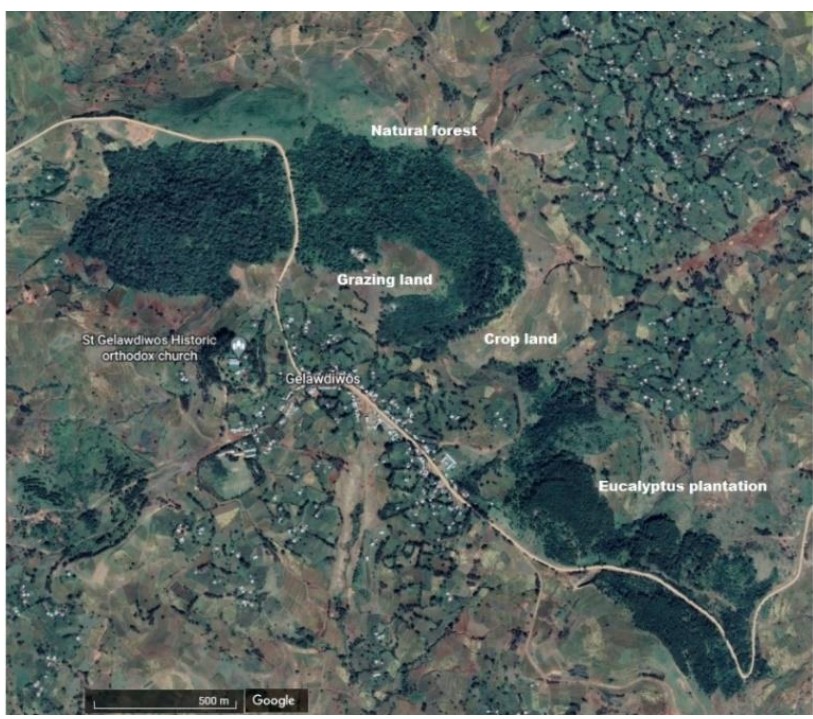

**Figure A1.** Map of the site at Gelawdios, Amhara, Northwestern Ethiopia (11°38′25″ N and 37°48′55″ E), showing the four adjacent land-use systems (natural forest, eucalyptus plantation, cropland, and grazing land). Based on Figure 1 of [12].

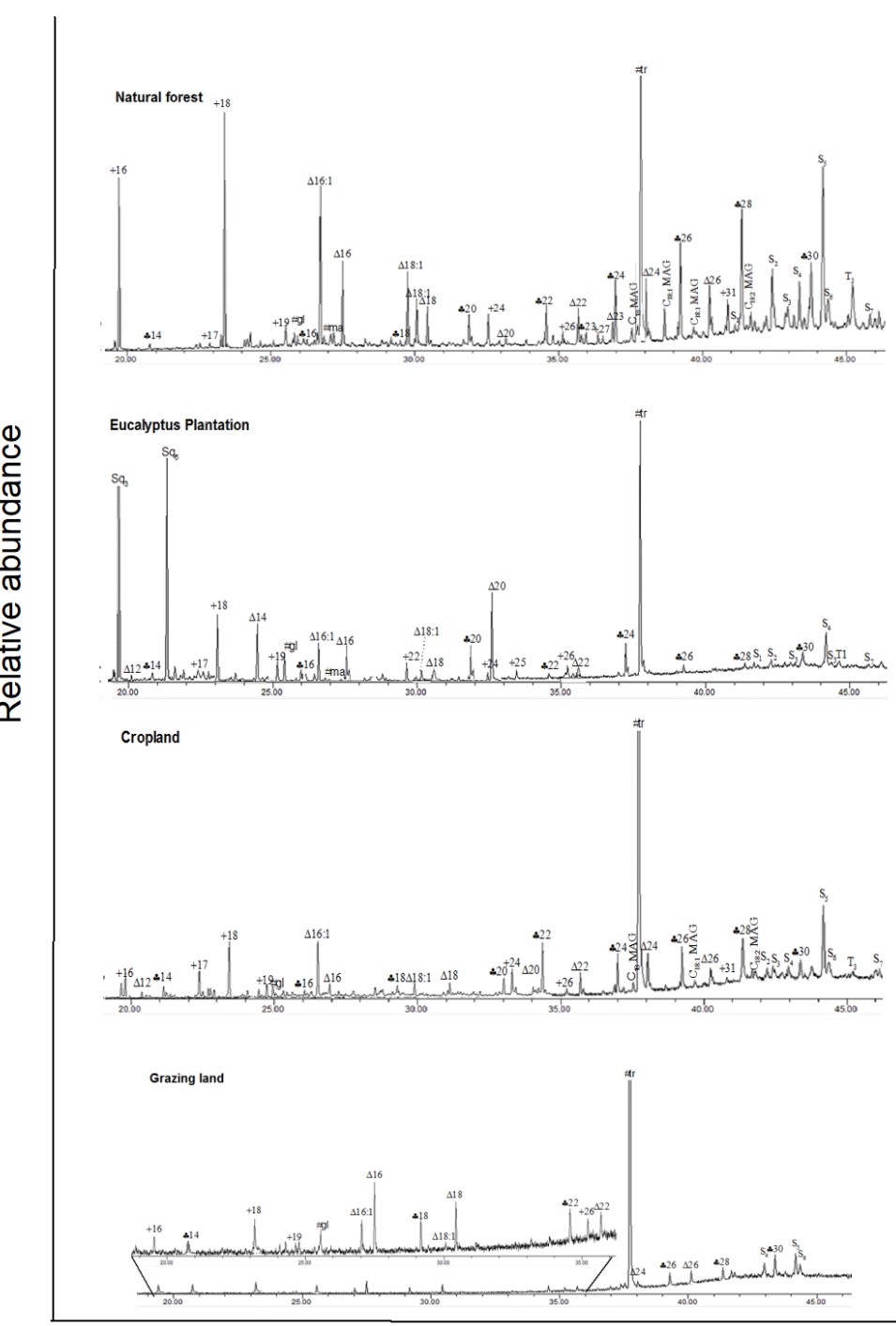

**Figure A2.** Total ion GC–MS chromatograms (TIC) of the silylated solvent extracts of four land-use systems from Gelawdios, Ethiopia. Notes: ♣, *n*-alkanols; +, *n*-alkanes; Δ, *n*-alkanoic acids; #, carbohydrates (gl, glucose; ma, mannose; su, sucrose); MAG, monoacylglycerides; $S_1$–$S_7$, steroids; $T_1$–$T_5$, triterpenoids; $U_1$, unidentified. Numbers refer to total carbon numbers in aliphatic lipid series. Detailed description of each compound with its quantity, molecular formula, and molecular weight in Appendix A Table A1.

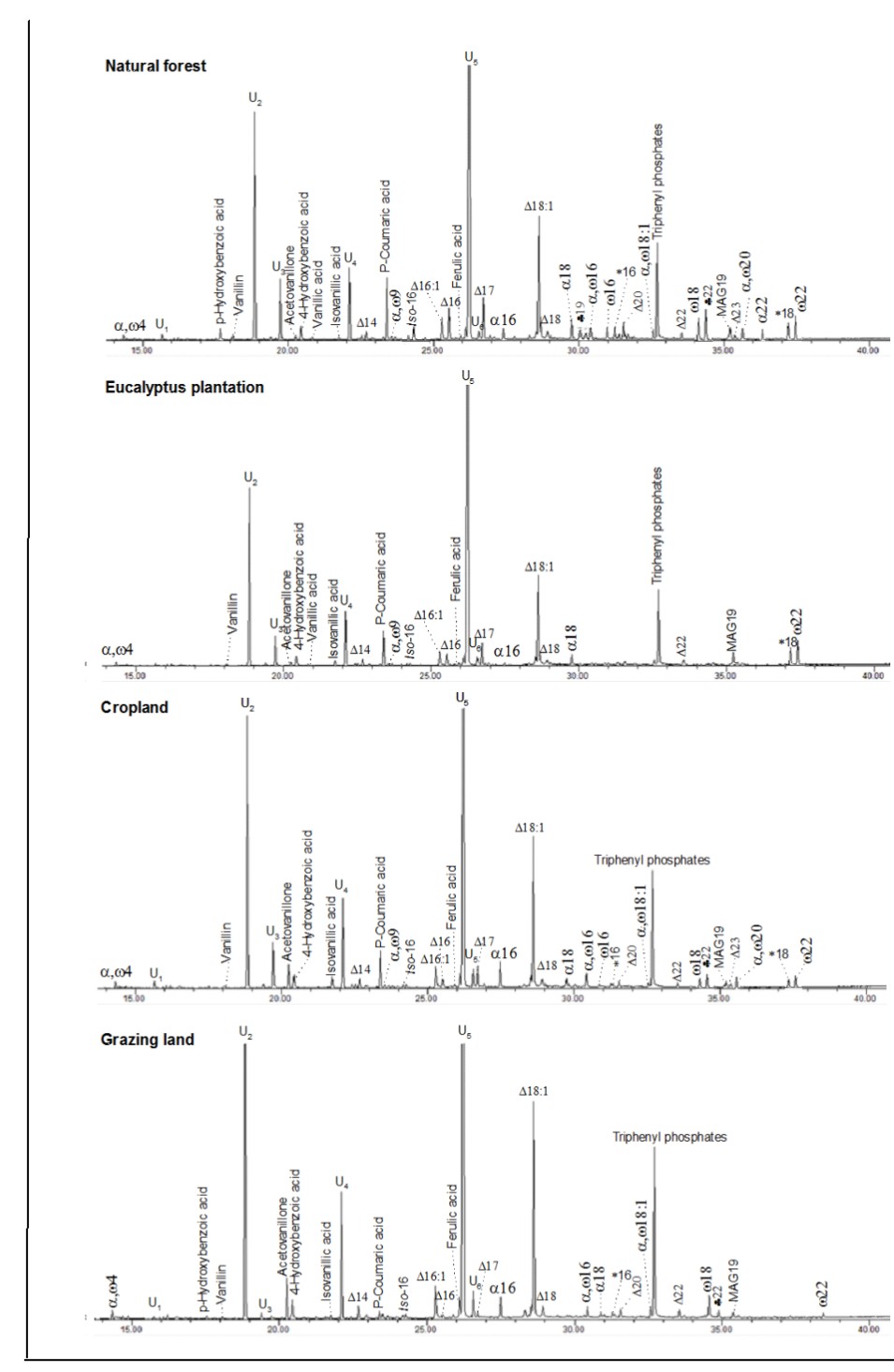

**Figure A3.** Total ion GC–MS chromatograms (TIC) of the methylated and silylated extracts after base hydrolysis of four land-use systems from Gelawdios, Ethiopia. Notes: ♣, *n*-alkanols; +, *n*-alkanes; Δ, *n*-alkanoic acids; iso-alkanoic acids; α-alkanoic acids; α,ω-alkanedioic acids; ω-hydroxyalkanoic acids; * Mid chain substituted hydroxy acids; steroids (ß-sitpsterol); organophosphate; phenols; and $U_1$–$U_5$, unidentified. Numbers refer to total carbon numbers in aliphatic lipid series. Detailed description of each compound, with its quantity, molecular formula, and molecular weight, is in Appendix A Table A2.

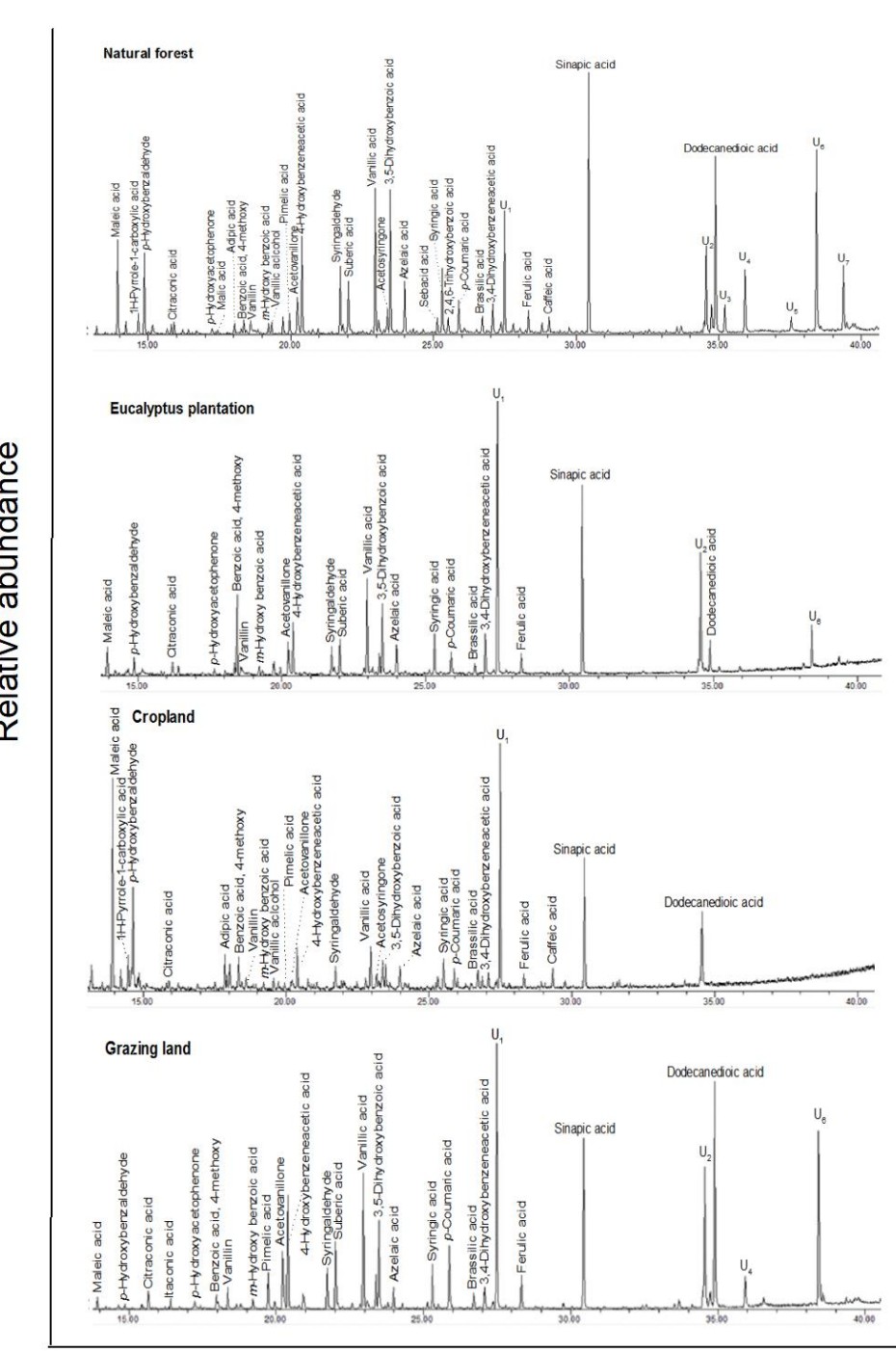

**Figure A4.** Total ion GC–MS chromatograms (TIC) of the silylated CuO oxidation products of four land-use systems from Gelawdios, Ethiopia. U$_1$–U$_7$, unidentified. Detailed description of each compound with its quantity, molecular formula, and molecular weight in Appendix A Table A3.

**Table A1.** Compounds ($\mu$g g$^{-1}$ C) identified in the solvent extracts of soil samples from different land-use systems in Gelawdios, Ethiopia. MF, molecular formula; MW, molecular weight. Values are mean $\pm$ SE; *n* =1–3 (see table footnotes).

| Compound [3] | MF | MW | Forest | Eucalyptus | Cropland | Grazing Land |
|---|---|---|---|---|---|---|
| *n-Alkanols* | | | | | | |
| 2-Tetradecanol | $C_{14}H_{30}O$ | 226 | | 65.3 $\pm$ 16.4 | 5.8 [1] | |
| 1-Hexadecanol | $C_{16}H_{34}O$ | 242 | | 43.3 [2] | 3.7 [1] | |
| 1-Octadecanol | $C_{18}H_{38}O$ | 270 | 2.1 [1] | | 6.0 [1] | 25.1 [1] |
| *n*-Eicosanol | $C_{20}H_{42}O$ | 298 | 9.2 [2] | 50.7 [2] | 5.1 [1] | |
| *n*-Docosanol | $C_{22}H_{46}O$ | 326 | 45.2 $\pm$ 25.4 | 96.8 $\pm$ 8.0 | 55.1 $\pm$ 10.2 | 46.9 $\pm$ 19.3 |
| 1-Tricosanol | $C_{23}H_{48}O$ | 340 | 6.7 [2] | | | 6.2 [1] |
| *n*-Tetracosanol | $C_{24}H_{50}O$ | 354 | 41.9 $\pm$ 29.9 | 80.2 $\pm$ 20.1 | 40.6 $\pm$ 18.1 | 69.8 $\pm$ 15.7 |
| *n*-Hexacosanol | $C_{26}H_{54}O$ | 382 | 84.0 $\pm$ 38.2 | 73.0 [2] | 47.4 $\pm$ 11.7 | 60.5 [2] |
| *n*-Octacosanol | $C_{28}H_{58}O$ | 410 | 93.7 $\pm$ 46.7 | 77.6 [2] | 86.0 $\pm$ 39.5 | 66.2 $\pm$ 14.8 |
| *n*-Tricontanol | $C_{30}H_{62}O$ | 438 | 41.4 [2] | 33.4 [1] | | 81.4 $\pm$ 15.3 |
| *Total Alkanols* | | | 324 $\pm$ 172 | 520 $\pm$ 212 | 250 $\pm$ 86 | 356 $\pm$ 103 |
| *n-Alkanes* | | | | | | |
| *n*-Heptadecane | $C_{17}H_{36}$ | 240 | | 162 $\pm$ 72 | 16.0 [2] | |
| Octadecane | $C_{18}H_{38}$ | 254 | 1.1 [1] | 58.6 [2] | | |
| Nonadecane | $C_{19}H_{40}$ | 268 | | 81.1 [2] | 8.6 [1] | |
| *n*-Docosane | $C_{22}H_{46}$ | 310 | | | 4.8 [1] | 3.5 [1] |
| Tricosane | $C_{23}H_{48}$ | 324 | | | 4.0 [1] | 3.1 [1] |
| *n*-Tetracosane | $C_{24}H_{50}$ | 338 | 6.2 [2] | | | |
| *n*-Pentacosane | $C_{25}H_{52}$ | 352 | | 16.9 [1] | 18.1 $\pm$ 0.8 | 11.7 [2] |
| *n*-Hexacosane | $C_{26}H_{54}$ | 366 | | | 19.6 [1] | 29.6 $\pm$ 8.2 |
| *n*-Heptacosane | $C_{27}H_{56}$ | 380 | 201 $\pm$ 76 | 45.0 [1] | 39.9 [1] | 1.9 [1] |
| Hentriacontane | $C_{31}H_{64}$ | 436 | 13.6 [2] | 27.2 [1] | 27.6 [1] | 8.0 [1] |
| *Total Alkanes* | | | 222 $\pm$ 71 | 391 $\pm$ 78 | 139 $\pm$ 66 | 58 $\pm$ 15 |
| *n-Alkanoic Acids* | | | | | | |
| Nonanoic acid | $C_9H_{18}O_2$ | 158 | 8.7 [2] | 14.5 [1] | 38.7 $\pm$ 6.2 | 37.6 $\pm$ 13.1 |
| Dodecanoic acid | $C_{12}H_{24}O_2$ | 200 | | 188.3 $\pm$ 33.6 | 19.4 [2] | 10.9 [2] |
| *n*-Tetradecanoic acid | $C_{14}H_{28}O_2$ | 228 | 10.0 [2] | 149.3 $\pm$ 18.5 | 4.3 [1] | 11.2 [1] |
| *n*-Hexadecanoic acid | $C_{16}H_{32}O_2$ | 256 | 48.9 $\pm$ 25.4 | 169.8 $\pm$ 22.1 | 63.9 $\pm$ 22.0 | 34.0 $\pm$ 2.4 |
| n-Octadecanoic acid (18:1) | $C_{18}H_{34}O_2$ | 282 | 20.5 [2] | 132.4 $\pm$ 17.3 | | 24.9 [2] |
| 9-Octadecanoic acid (Z) | $C_{18}H_{34}O_2$ | 282 | 10.3 [2] | 89.9 $\pm$ 20.9 | | 12.5 [2] |
| *n*-Octadecanoic acid | $C_{18}H_{36}O_2$ | 284 | 21.1 $\pm$ 7.5 | 114.7 $\pm$ 41.7 | 38.4 $\pm$ 14.0 | 48.4 $\pm$ 23.8 |
| *n*-Eicosanoic acid | $C_{20}H_{40}O_2$ | 312 | 0.4 [1] | 14.8 [1] | | 3.0 [1] |
| *n*-Docosanoic acid | $C_{22}H_{44}O_2$ | 340 | 132.0 [2] | 131.9 $\pm$ 44.9 | 25.8 $\pm$ 9.9 | 18.8 $\pm$ 1.3 |
| Tricosanoic acid | $C_{23}H_{46}O_2$ | 354 | 11.8 [2] | | | |
| *n*-Tetracosanoic acid | $C_{24}H_{48}O_2$ | 368 | 75.5 $\pm$ 46.0 | 81.3 [2] | 11.6 [2] | 27.2 $\pm$ 2.9 |
| *n*-Hexacosanoic acid | $C_{26}H_{52}O_2$ | 396 | 60.3 $\pm$ 14.0 | 42.5 [1] | 5.2 [1] | 43.8 $\pm$ 19.4 |
| *Total Alkanoic Acids* | | | 449 $\pm$ 184 | 1129 $\pm$ 128 | 207 $\pm$ 50 | 272 $\pm$ 31 |

**Table A1.** *Cont.*

| Compound [3] | MF | MW | Forest | Eucalyptus | Cropland | Grazing Land |
|---|---|---|---|---|---|---|
| *Iso-Alkanoic Acids* | | | | | | |
| Iso-heptadecanoic acid | $C_{17}H_{34}O_2$ | 270 | 24.5 [2] | | | |
| *Monoacylglycerides* | | | | | | |
| (±)-2,3-Dihydroxypropyl hexadecanoate (C16) | $C_{19}H_{38}O_4$ | 330 | 9.2 [2] | 19.1 [1] | 37.1 [2] | |
| Monostearin (C18:1) | $C_{21}H_{42}O_4$ | 358 | 16.1 [2] | 134.4 ± 26.1 | 97.0 ± 19.14 | 64.9 ± 8.9 |
| Linolein, 1-mono (C18:1) | $C_{21}H_{38}O_4$ | 354 | 20.6 ± 9.5 | 31.9 [1] | | 9.3 [1] |
| 2-Monolinolenin (C18:2) | $C_{21}H36O_4$ | 352 | 6.4 [1] | 74.3 ± 16.0 | | 9.7 [1] |
| *Total Monoacylglycerides* | | | 52 ± 26 | 260 ± 19 | 134 ± 46 | 84 ± 2 |
| *Carbohydrates* | | | | | | |
| D-Glucose | $C_6H_{12}O_6$ | 180 | 16.2 [2] | 90.1 ± 32.9 | 37.8 ± 16.6 | 46.7 ± 14.7 |
| Mannose | $C_6H_{12}O_6$ | 180 | 16.9 ± 7.0 | 212 ± 78.4 | 78.4 ± 25.3 | 46.0 ± 14.0 |
| Sucrose | $C_{12}H_{22}O_{11}$ | 342 | 7.7 [2] | 19.6 [1] | 19.5 [2] | 33.0 ± 11.3 |
| Trehalose | $C_{12}H_{22}O_{11}$ | 342 | 3895 ± 712 [1] | 2107 ± 217 [2] | 1684 ± 191 [2] | 1768 ± 157 [2] |
| *Total Carbohydrates* | | | 3936 ± 703 | 2429 ± 256 | 18120 ± 229 | 1894 ± 192 |
| *Steroids and Terpenoids* | | | | | | |
| Aromadendrene | $C_{15}H_{24}$ | 204 | 0.5 [1] | 125 [2] | 13.8 [1] | |
| γ-Elemene | $C_{15}H_{24}$ | 204 | | 183.0 ± 81.1 | 13.5 [1] | |
| Ledol | $C_{15}H_{26}O$ | 222 | | 157.2 ± 32.9 | 9.3 [1] | |
| (-)-Globulol | $C_{15}H_{26}O$ | 222 | | 72.0 ± 7.5 | 17.1 [2] | 11.8 [1] |
| trans-Farnesol | $C_{15}H_{26}O$ | 222 | | 753 ± 64.9 | 48.2 [1] | |
| Cholesterol | $C_{27}H_{46}O$ | 386 | 11.4 [2] | 98.8 ± 33.0 | 62.2 [2] | 23.6 [2] |
| Ergostrol | $C_{28}H_{44}O$ | 396 | 36.8 [2] | 94.7 [1] | | |
| Campesterol | $C_{28}H_{48}O$ | 400 | 32.9 [2] | 117.3 ± 28.7 | 87.9 [2] | 104.7 ± 0.3 |
| Stigmasterol | $C_{29}H_{48}O$ | 412 | 56.0 [2] | 243.1 ± 30.4 | 110.2 ± 17.4 | 138.9 ± 18.3 |
| ß-Sitosterol | $C_{29}H_{50}O$ | 414 | 232.2 ± 93.8 | 243.2 ± 6.0 | 237.9 ± 46.1 | 142.9 ± 25.7 |
| Stigmastanol | $C_{29}H_{52}O$ | 416 | | | | 119.4 ± 34.2 |
| ß-Amyrin | $C_{30}H_{50}O$ | 426 | 81.2 [2] | 161.8 ± 1.8 | 98.9 ± 16.3 | |
| α-Amyrin | $C_{30}H_{50}O$ | 426 | 14.7 [2] | 66.1 ± 17.9 | | |
| Lupeol | $C_{30}H_{50}O$ | 426 | 9.7 [1] | 87.3 [2] | 13.3 [1] | 28.0 [1] |
| Sitosterone | $C_{29}H_{48}O$ | 412 | 13.3 [1] | 73.6 [2] | 17.0 [1] | |
| Erthrodiol | $C_{30}H_{50}O_2$ | 442 | 47.4 [2] | 343.6 ± 100.3 | | |
| Oleanolic acid | $C_{30}H_{48}O_3$ | 456 | 85.4 ± 52.5 | 642.1 ± 132.5 | | 46.9 [2] |
| *Total Steroids and Terpenoids* | | | 622 ± 267 | 3461 ± 341 | 729 ± 80 | 616 ± 616 |
| Unidentified | | | 37.6 ± 3.4 | 126.2 ± 10.8 | 238.1 ± 99.6 | 67.7 [2] |
| Total aliphatic lipids | | | 5007 ± 746 | 4729 ± 531 | 2550 ± 430 | 2664 ± 300 |
| Total solvent extracts | | | 5629 ± 740 | 8190 ± 857 | 3279 ± 508 | 3280 ± 394 |

[1] Detected only in one sample. [2] Detected only in two samples. [3] All polar compounds were identified as their trimethylsilyl (TMS) derivatives.

**Table A2.** Compounds (µg g$^{-1}$ C) with molecular formula (MF) and molecular weight (MW) identified from base hydrolyzed of soil samples from different land-use systems in Gelawdios, Ethiopia. MF, molecular formula; MW, molecular weight. Values are mean ± SE, *n* = 1–3 (see below footnotes). All values as µg g$^{-1}$ C, except triphenyl phosphate (mg g$^{-1}$ C).

| Compound | MF | MW | Forest | Eucalyptus | Cropland | Grazing Land |
|---|---|---|---|---|---|---|
| *n-Alkanols* [3] | | | | | | |
| *n*-Hexadecanol | $C_{16}H_{34}O$ | 242 | 7.4 [1] | 20.2 | 66.4 [2] | 75.3 [2] |
| *n*-Octadecanol | $C_{18}H_{38}O$ | 270 | 9.1 [1] | 10.6 | 10.8 [1] | |
| *n*-Nonadecanol | $C_{19}H_{40}O$ | 284 | 40.5 ± 10.5 | 85.6 ± 12.9 | 80.7 [2] | |
| *n*-Docosanol | $C_{22}H_{46}O$ | 326 | 47.1 ± 5.5 | 121 ± 8 | 129 ± 36 | 552 ± 324 |
| *n*-Tetracosanol-1 | $C_{24}H_{50}O$ | 354 | | 13.0 [1] | | |
| *n*-Hexacosanol | $C_{26}H_{54}O$ | 382 | | 14.6 [1] | 14.3 [1] | |
| *n*-Octacosanol | $C_{28}H_{58}O$ | 410 | | | 13.0 [1] | |
| *Total n-Alkanols* | | | 104 ± 21 | 265 ± 12 | 314 ± 21 | 627 ± 362 |
| *n-Alkanoic Acids* | | | | | | |
| *n*-Tetradecanoic acid | $C_{14}H_{28}O_2$ | 228 | 128 ± 16 | 352 ± 20.2 | 540 ± 118 | 362.0 ± 92 |
| *n*-Hexadecanoic acid (C16:1) | $C_{16}H_{30}O_2$ | 254 | 390 ± 50 | 1030 ± 40 | 969 ± 18 | 1569 ± 127 |
| *n*-Hexacosanoic acid | $C_{16}H_{32}O_2$ | 256 | 579 ± 301 | 11.5 ± 49.1 | 651.2 ± 17.6 | 128 ± 70 |
| *n*-Heptadecanoic acid | $C_{17}H_{34}O_2$ | 270 | 472 ± 65.8 | 514 ± 27.4 | 592 ± 82.7 | 62.3 ± 57.0 |
| *n*-Octadecanoic acid, cis-9 (C18:1) | $C_{18}H_{34}O_2$ | 282 | 54.6 ± 12.2 | 106 ± 11 | 193 ± 38 | 321 ± 78 |
| *n*-Octadecanoic acid (C18:1) | $C_{18}H_{34}O_2$ | 282 | 2710 ± 790 | 6095 ± 118 | 6631 ± 437.1 | 8089 ± 655 |
| *n*-Octadecanoic acid | $C_{18}H_{36}O_2$ | 284 | 187 ± 40.1 | 408 ± 5 | 621 ± 12.4 | 661 ± 53 |
| *n*-Nonadecanoic acid | $C_{19}H_{38}O_2$ | 298 | 22.7 [2] | 41.9 [2] | | |
| *n*-Eicosanoic acid | $C_{20}H_{40}O_2$ | 312 | 83.2 ± 13.4 | 213 ± 6.7 | 322.6 ± 74.7 | 347 ± 36 |
| *n*-Docosanoic acid | $C_{22}H_{44}O_2$ | 340 | 93.3 ± 28.1 | 324 ± 61 | 292 ± 108 | 342 ± 76 |
| *n*-Triacosanoic acid | $C_{23}H_{46}O_2$ | 354 | 47.8 ± 4.1 | 139 ± 9.3 | 204 ± 52.2 | 206 ± 26.8 |
| *n*-Tetracosanoic acid | $C_{24}H_{48}O_2$ | 368 | 7.6 [1] | 26.9 [1] | | 57.7 [2] |
| *n*-Hexacosanoic acid | $C_{26}H_{52}O_2$ | 396 | 5.5 [1] | 14.6 [1] | | |
| *n*-Octacosanoic acid | $C_{28}H_{56}O_2$ | 424 | 7.3 [1] | 21.3 [1] | 46.1 [1] | |
| *n*-Triacontanoic acid | $C_{30}H_{60}O_2$ | 302 | | | | 11.0 [1] |
| *Total Alkanoic Acids* | | | 4788 ± 1180 | 9998 ± 179 | 11063 ± 452 | 12157 ± 1173 |
| *α-Alkanoic Acids* | | | | | | |
| α-Hydroxydocosanoic acid | $C_{22}H_{44}O_3$ | 356 | 17.4 [2] | 12.5 [1] | | |
| α-Hydroxytetracosanoic acid | $C_{24}H_{48}O_3$ | 384 | | | | 75.6 [2] |
| α-hydroxypentacosanoic acid | $C_{25}H_{50}O_3$ | 398 | | | | 24.5 [1] |
| *Total α-Alkanoic Acids* | | | 364 ± 60 | 515 ± 124 | 576 ± 100 | 572 ± 357 |
| *α,ω-Alkanedioic Acids* | | | | | | |
| α,ω-Butanedioic acid | $C_4H_6O_4$ | 118 | 62.2 ± 20.8 | 123.7 ± 5.2 | 60.3 ± 14.3 | 178.7 ± 67.9 |
| α,ω-Nonanedioic acid | $C_9H_{16}O_4$ | 188 | 39.0 ± 12.0 | 94.5 ± 4.2 | 109.2 ± 41.7 | 93.4 ± 24.4 |

**Table A2.** *Cont.*

| Compound | MF | MW | Forest | Eucalyptus | Cropland | Grazing Land |
|---|---|---|---|---|---|---|
| α,ω-Hexadecanedioic acid | $C_{16}H_{30}O_4$ | 286 | 64.4 ± 9.1 | 110.6 ± 23.3 | 181.3 ± 42.3 | 248.4 ± 9.1 |
| α,ω-Octadec-9-enedioic acid (C18:1) | $C_{18}H_{32}O_4$ | 312 | 120.0 ± 18.5 | 151.8 ± 52.7 | 93.7 [2] | |
| α,ω-Eicosanedioc acid | $C_{20}H_{38}O_4$ | 342 | 56.5 ± 11.7 | 161.4 ± 9.4 | 144.4 ± 15.6 | 157.8 ± 17.9 |
| *Total α,ω-Alkanedioic Acids* | | | 342 ± 49 | 642 ± 41 | 589 ± 125 | 678 ± 70 |
| *ω-Hydroxyalkanoic Acids* [3] | | | | | | |
| ω-Hydroxyhexadecanoic acid | $C_{16}H_{32}O_3$ | 272 | 97.5 ± 27.7 | 81.4 [2] | 137.5 [2] | 124.6 [2] |
| ω-Hydroxyoctadecanoic acid | $C_{18}H_{34}O_3$ | 298 | 41.9 ± 4.4 | 98.4 ± 13.7 | 103.9 ± 48.2 | 8.4 [1] |
| ω-Hydroxydocosanoic acid | $C_{22}H_{44}O_3$ | 358 | 83.1 ± 19.2 | 125 ± 28 | 106 ± 30 | 10.8 [1] |
| ω-Hydroxyoctacosanoic acid | $C_{28}H_{56}O_3$ | 440 | 136 ± 72 | 235 ± 35 | 478 ± 43 | 75.2 [1] |
| ω-Hydroxytriacontanoic acid | $C_{30}H_{60}O_3$ | 468 | 377 ± 70 | 565.2 ± 13.9 | 518.7 ± 69.9 | 78.7 [1] |
| *Total ω-Hydroxyalkanoic Acids* | | | 736 ± 31 | 1105 ± 26 | 1344 ± 30 | 298 ± 141 |
| *Mid-Chain Substituted Hydroxy Acids* | | | | | | |
| 10,16-Dihydroxyhexadecanoic | $C_{16}H_{32}O_4$ | 288 | 58.8 [2] | 152.8 [2] | 182.1 [2] | 223.3 [2] |
| 8-Hydroxyhexadecane-1,16-dioic acid | $C_{16}H_{30}O_5$ | 302 | 67.4 ± 13.3 | 115 ± 26.6 | 123 ± 4.1 | |
| Dihydroxymethoxyoctadecanoic acid | $C_{18}H_{36}O_5$ | 332 | 3.1 [1] | 11.2 [1] | | |
| 9,10,18-Trihydroxyoctadecanoic acid | $C_{18}H_{36}O_5$ | 332 | 32.2 [1] | 90.9 [2] | 38.1 [2] | |
| 9,10-Dihydroxyoctadecane-1,18-dioic acid | $C_{19}H_{38}O_4$ | 330 | 65.5 ± 41.1 | 222 [2] | 78.3 [2] | 85.9 [2] |
| *Total Mid-Chain Substituted Hydroxy Acids* | | | 227 ± 111 | 592 ± 122 | 422 ± 160 | 309 ± 80 |
| Monoacylglycerides | $C_{19}H_{38}O_4$ | 330 | 238 ± 59 | 487 ± 128 | 432 ± 177 | |
| *Benzyles and Phenols* | | | | | | |
| *p*-Hydroxyacetophenone | $C_8H_8O_2$ | 136 | 21.7 ± 4.1 | 53.6 ± 5.7 | 93.0 ± 17.0 | 55.3 [2] |
| Vanillin | $C_8H_8O_3$ | 152 | 4.7 [1] | | | 38.0 [2] |
| Acetovanillone | $C_8H_{10}O_3$ | 154 | 51.4 [2] | 122 ± 32.7 | 137 [2] | 1080 ± 468 |
| 4-Hydroxybenzoic acid | $C_7H_6O_3$ | 138 | 204 ± 27 | 609 ± 34 | 692 ± 51 | 916 ± 116 |
| Vanillic acid | $C_8H_8O_4$ | 168 | 25.4 ± 10.7 | 74.3 ± 20.6 | 39.2 [2] | 53.0 [2] |
| Isovanillic acid | $C_8H_8O_4$ | 168 | 60.1 ± 5.1 | 217 ± 33 | 254 ± 71 | 40.1 [1] |
| *p*-Coumaric acid | $C_9H_8O_3$ | 164 | 19.1 [2] | 60.6 [2] | 66.7 [2] | 52.3 [1] |
| Ferulic acid | $C_{10}H_{10}O_4$ | 194 | 40.9 ± 5.1 | 96.8 ± 1.4 | 109 ± 29 | 67.5 [2] |
| *Total Benzyles and Phenols* | | | 428 ± 53 | 1233 ± 57 | 1390 ± 200 | 2303 ± 493 |

[1] Detected only in one sample. [2] Detected only in two samples; [3] *n*-alkohols, terpenols, and sterols were identified as TMS ethers.

**Table A3.** Major compounds ($\mu g\, g^{-1}$ C) identified in the CuO oxidation extracts of soil samples from different land-use systems in Gelawdios, Ethiopia. MF, molecular formula; MW, molecular weight. Values are mean $\pm$ SE; *n* = 1–3 (see below footnotes).

| Compound | MF | MW | Forest * | Eucalyptus | Cropland | Grazing Land |
|---|---|---|---|---|---|---|
| *Organophosphates* | | | | | | |
| Triphenyl phosphate (mg g$^{-1}$) | C$_{18}$H$_{15}$O$_4$P | 326 | 1.5 $\pm$ 0.2 | 4.5 $\pm$ 0.1 | 5.6 $\pm$ 0.3 | 6.8 $\pm$ 0.6 |
| *Hydroxy Benzene Products* | | | | | | |
| p-Hydroxyacetophenone | C$_8$H$_8$O$_2$ | 136 | 198 | 145 [1] | nb | 483 [2] |
| Benzoic acid, 4-methoxy-/*p*-Anisic acid | C$_8$H$_8$O$_3$ | 152 | 524 | 233 [1] | 301 [1] | 537 [2] |
| *m*-Hydroxy benzoic acid | C$_7$H$_6$O$_3$ | 138 | 271 | 509 $\pm$ 61 | 625 $\pm$ 2 | 241 [1] |
| 3,5-dihydroxybenzoic acid | C$_7$H$_6$O$_4$ | 154 | 2991 | 665 $\pm$ 119 | 469 [2] | 360 [1] |
| 2,4,6-Trihydroxybenzoic acid | C$_7$H$_6$O$_5$ | 170 | 181 | 442 $\pm$ 27 | 247 [1] | 705 $\pm$ 145 |
| 3,4-Dihydroxybenzeneacetic acid | C$_8$H$_8$O$_4$ | 168 | 592 | 801 $\pm$ 180 | 432 [2] | 751 $\pm$ 214 |
| *Total Hydroxy Benzene Products* | | | 31218 | 2795 $\pm$ 413 | 2074 $\pm$ 417 | 3078 $\pm$ 910 |
| *Protein and Polysaccharide Products* | | | | | | |
| Benzoic acid | C$_7$H$_6$O$_2$ | 122 | 626 | | | |
| Butan-1,4-dioic acid/succinic acid | C$_4$H$_6$O$_4$ | 118 | 2584 | 420 [2] | 642 $\pm$ 15 | 242 [1] |
| 2-Butenedioic acid/fumaric acid | C$_4$H$_4$O$_4$ | 116 | 126 | 400 [2] | 840 $\pm$ 55 | 175 [1] |
| Maleic acid | C$_4$H$_4$O$_4$ | 116 | 1591 | 183 [1] | 519 [1] | 237 [1] |
| 1H-pyrrole-1-carboxylic acid | C$_5$H$_5$NO$_2$ | 111 | 902 | 532 $\pm$ 64 | 658 $\pm$ 62 | 196 [1] |
| *p*-Hydroxybenzaldehyde | C$_7$H$_6$O$_2$ | 122 | 997 | 365 [2] | 213 [1] | 203 [1] |
| 4-Hydroxybenzeneacetic acid | C$_8$H$_8$O$_3$ | 152 | 1449 | 837 $\pm$ 183 | 742 $\pm$ 74 | 554 $\pm$ 35 |
| 2-Buten-1,4-dioic acid/citraconic acid | C$_5$H$_6$O$_4$ | 130 | 165 | 151 [1] | | 200 [1] |
| Penten-1,5-dioic acid/itaconic acid | C$_5$H$_6$O$_4$ | 130 | 121 | 499 $\pm$ 44 | 488 [2] | 375 [2] |
| Hydroxybutan-1,4-dioic acid/Malic acid | C$_4$H$_6$O$_5$ | 134 | 150 | 301 [2] | 425 [2] | 518 $\pm$ 19 |
| *Total Protein and Polysaccharide Products* | | | 8713 | 3686 | 4528 | 2700 |
| *Lignin Monomers* | | | | | | |
| Vanillin/vanillaldehyde | C$_8$H$_8$O$_3$ | 152 | 564 | 555 $\pm$ 78 | 219 [1] | 576 $\pm$ 36 |
| Vanillic alcohol | C$_8$H$_{10}$O$_3$ | 154 | 188 | 292 [2] | nd | nd |
| Acetovanillone | C$_8$H$_{10}$O$_3$ | 154 | 573 | 744 $\pm$ 155 | 654 $\pm$ 35 | 587 [2] |
| Vanillic acid | C$_8$H$_8$O$_4$ | 168 | 1169 | 178 [1] | 725 $\pm$ 110 | |
| *Total Vanillyls* | | | 2493 | 1768 $\pm$ 353 | 1598 $\pm$ 364 | 1163 $\pm$ 384 |
| Syringaldehyde | C$_9$H$_{10}$O$_4$ | 182 | 272 | 177 [1] | 252 [1] | 389 [1] |
| Acetosyringone | C$_{10}$H$_{12}$O$_4$ | 196 | 491 | 576 $\pm$ 89 | 707 $\pm$ 52 | 1091 $\pm$ 221 |
| Syringic acid | C$_9$H$_{10}$O$_5$ | 198 | 619 | 741 $\pm$ 157 | 617 $\pm$ 36 | 199 [1] |
| *Total Syringyls* | | | 1382 | 1494 $\pm$ 70 | 1576 $\pm$ 339 | 1827 $\pm$ 691 |
| *Cinnamyls* | | | | | | |
| P-Coumaric acid | C$_9$H$_8$O$_3$ | 164 | 299 | 556 $\pm$ 72 | 660 $\pm$ 62 | 253 [2] |
| Ferulic acid | C$_{10}$H$_{10}$O$_4$ | 194 | 267 | 611 $\pm$ 83 | 656 $\pm$ 29 | 1015 $\pm$ 271 |
| Caffeic acid | C$_9$H$_8$O$_4$ | 180 | 388 | 139 [1] | | 336 [2] |
| Sinapic acid | C$_{11}$H$_{12}$O$_5$ | 284 | 4132 [1] | 941 $\pm$ 295 [2] | 932 $\pm$ 87 [2] | 973 $\pm$ 145 [2] |

**Table A3.** *Cont.*

| Compound | MF | MW | Forest * | Eucalyptus | Cropland | Grazing Land |
|---|---|---|---|---|---|---|
| *Total Cinnamyls* | | | 5086 | 2247 ± 339 | 2249 ± 91 | 2577 ± 179 |
| *Total Benzyles and Phenols* | | | 48892 | 11991 ± 1394 | 12025 ± 1693 | 11344 ± 3471 |
| *Dicarboxylic Acids* | | | | | | |
| Adipic acid | $C_6H_{10}O_4$ | 146 | 439 | 603 [2] | 863 [2] | 167 [1] |
| Pimelic acid | $C_7H_{12}O_4$ | 160 | 303 | 303 [2] | 402 [2] | 254 [1] |
| Suberic acid | $C_8H_{14}O_4$ | 174 | 818 | 708 ± 165 | 706 ± 108 | 760 [2] |
| Azelaic acid | $C_9H_{16}O_4$ | 118 | 1120 | 893 ± 214 | 672 ± 53 | 451 [2] |
| Sebacic acid | $C_{10}H_{18}O_4$ | 202 | 294 | 147 [1] | | 217 [1] |
| Brassilic acid | $C_{11}H_{20}O_4$ | 216 | 292 | 527 ± 66 | 657 ± 44 | 790 ± 114 |
| Dodecanedioic acid | $C_{12}H_{22}O_4$ | 230 | 1802 | 753 ± 176 | 469 [2] | 1357 ± 240 |
| *Total Dicarboxylic Acids* | | | 5069 | 3933 ± 995 | 3769 ± 362 | 3995 ± 911 |
| *Identified Total CuO Products* | | | 53961 | 15924 ± 1700 | 15794 ± 1278 | 15340 ± 4044 |
| *Unidentified* | | | 8475 | 5425 ± 507 | 5638 ± 258 | 6250 ± 1147 |
| *Total CuO Products* | | | 62436 | 21349 ± 507 | 21432 ± 258 | 21590 ± 1147 |

* The third sample in the forest was contaminated during extraction. Therefore, all values in CuO oxidation product of forest soil are an average of two samples. [1] Values detected only from one sample; [2] values detected only from two samples.

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
