# Peer review of "The Biological Origins of Soil Organic Matter in Different Land-Uses in the Highlands of Ethiopia"

_forests, doi:10.3390/f13040560_

Round 1

Reviewer 1 Report

Congratulations to the authors, excellent work. It shows a methodological strategy still little known for understanding soil carbon. Below are some suggestions for improving the approach.
-describe better and further characterize the different land uses, especially information about the land. Do you have soil fertility and physics analysis? if yes it would be relevant.
-It is important to reference and present the general pattern of the relationship between contribution of woody and non-woody angiosperms, how can we compare?
- the statistic is questionable, since the ANOVA assumptions are probably not met. I suggest using other analysis of variance models for comparison. For better results, it would also be relevant to have some multivariate analyzes that show the patterns between C/N, molecules and different land uses, it would be very enriching.

Reviewer 2 Report

Although the manuscript seems to be interesting and appropriate for the Journal, it may be improved. Overall the manuscript was written and structured appropriately.
There is no reference to the research field in the introduction, and it would be beneficial to emphasize the importance of this research. Additionally, it is worth including a figure to show the study area, different land uses, and soil sampling points. For more comments, please see the annotated attached pdf.

Reviewer 3 Report

The article is interesting and well written.
I recommend adding recent bibliographic titles to the Introduction. The time interval in which the research was performed must be entered in the Materials and Methods. Conclusions must be supported by results (concrete values ​​for different indicators)
